# Large protein complex interfaces have evolved to promote cotranslational assembly

**Mihaly Badonyi, Joseph A Marsh***

MRC Human Genetics Unit, Institute of Genetics and Cancer, University of Edinburgh, Edinburgh, United Kingdom

**Abstract** Assembly pathways of protein complexes should be precise and efficient to minimise misfolding and unwanted interactions with other proteins in the cell. One way to achieve this efficiency is by seeding assembly pathways during translation via the cotranslational assembly of subunits. While recent evidence suggests that such cotranslational assembly is widespread, little is known about the properties of protein complexes associated with the phenomenon. Here, using a combination of proteome-specific protein complex structures and publicly available ribosome profiling data, we show that cotranslational assembly is particularly common between subunits that form large intermolecular interfaces. To test whether large interfaces have evolved to promote cotranslational assembly, as opposed to cotranslational assembly being a non-adaptive consequence of large interfaces, we compared the sizes of first and last translated interfaces of heteromeric subunits in bacterial, yeast, and human complexes. When considering all together, we observe the N-terminal interface to be larger than the C-terminal interface 54% of the time, increasing to 64% when we exclude subunits with only small interfaces, which are unlikely to cotranslationally assemble. This strongly suggests that large interfaces have evolved as a means to maximise the chance of successful cotranslational subunit binding.

***For correspondence:**
joseph.marsh@ed.ac.uk

**Competing interest:** The authors declare that no competing interests exist.

## Editor's evaluation

The authors use a combination of proteome-specific protein complex structures and publicly available ribosome profiling data to show that cotranslational assembly is favored by large N-terminal intermolecular interfaces. The manuscript represents an important contribution to the field of protein biosynthesis pathways by suggesting an intuitive evolutionary mechanism that can promote co-translational assembly pathways in mammalians, yeast, and bacteria.

## Introduction

The majority of proteins across all domains of life function as part of multimeric complexes. Although we have a comprehensive understanding of the diverse quaternary structure space occupied by complexes (*Ahnert et al., 2015*), much less is known about where, when, and how their component subunits assemble. Continuing advances in cryo-electron microscopy (*Fontana et al., 2022*), mass photometry (*Soltermann et al., 2020*), and genetic interaction mapping (*Braberg et al., 2020*) are facilitating a transition towards a structural view of proteomes (*Levy and Vogel, 2021*). While, at the present, our structural analyses are primarily limited to complexes in the Protein Data Bank (*Berman et al., 2000*), there is a new generation of multiscale protein complex modelling approaches (*Evans et al., 2021*; *Humphreys et al., 2021*; *Fontana et al., 2022*; *Gao et al., 2022*) promising to fill the gap and accelerate structure-based discovery. Our understanding of proteomes has also been

dramatically improved by the development of ribosome profiling, which has provided us with quantitative measurements at the level of translation. Alterations of the technique revealed the cotranslational action of chaperones (*Oh et al., 2011*; *Shiber et al., 2018*; *Stein et al., 2019*), shed light on the role of collided ribosomes in proteostasis (*Arpat et al., 2020*; *Han et al., 2020*; *Zhao et al., 2021*), and supported the view of the ribosome as a signalling hub (*D'Orazio and Green, 2021*). To the present work, however, it is of outstanding relevance that ribosome profiling has laid down strong evidence that the assembly of protein complexes often starts on the ribosome (*Duncan and Mata, 2011*; *Natan et al., 2017*; *Sepulveda et al., 2018*; *Shiber et al., 2018*; *Kamenova et al., 2019*; *Panasenko et al., 2019*; *Fujiwara et al., 2020*; *Bertolini et al., 2021*; *Seidel et al., 2022*).

Two factors appear to be particularly important for cotranslational assembly: the proximity of nascent chains on adjacent (*cis*) or between juxtaposed (*trans*) ribosomes, and the localisation of interface residues towards the N-terminus of a protein, which allows more time for an interaction to occur during translation (*Shieh et al., 2015*; *Natan et al., 2018*; *Kamenova et al., 2019*). Recent findings demonstrated that homomers, formed from multiple copies of a single type of polypeptide chain, frequently assemble on the same transcript via the interaction of adjacent elongating ribosomes (*Bertolini et al., 2021*). This mechanism is highly effective because it takes advantage of the fact that homomeric subunits are identical; therefore nascent chains are essentially colocalised by the nature of their synthesis. Although homomers may benefit from polysome-driven assembly, it requires allocation of cellular resources to ensure at least two ribosomes are actively translating the same mRNA at any one time (*Liu et al., 2016a*). On the other hand, heteromers, products of different genes that physically interact, can only employ the *trans* mode of assembly in eukaryotes, providing mechanisms that colocalise their transcripts exist (*Liu et al., 2016b*; *Pizzinga et al., 2019*; *Wang et al., 2020*; *Chen and Mayr, 2022*). In contrast to homomers, cotranslational assembly of heteromers may only require a single ribosome on each mRNA, which could allow lowly abundant regulatory proteins to cotranslationally assemble (*Heyer and Moore, 2016*; *Biever et al., 2020*). Alternate ribosome usage and translation-coupled assembly can explain how cells achieve efficient construction of complexes with uneven stoichiometry, accounting for a substantial fraction of heteromeric complexes (*Marsh et al., 2015*).

Despite growing evidence supporting the importance of cotranslational assembly, far less is known about the properties of the interfaces involved. It has been observed that cotranslationally binding subunits have a tendency to fall out of solution or become degraded by orphan subunit surveillance mechanisms in the absence of their partner subunits (*Choe et al., 2016*; *Juszkiewicz and Hegde, 2018*; *Natan et al., 2018*; *Shiber et al., 2018*; *Kamenova et al., 2019*). This observation may be explained under two assumptions: N-terminal interfaces are aggregation prone due to interference with cotranslational folding (*Ciryam et al., 2013*; *Jacobs and Shakhnovich, 2017*; *Kudva et al., 2018*; *Kramer et al., 2019*), and/or that cotranslationally forming interfaces possess unique structural properties that predispose them to aggregation in the absence of binding partners. Whilst there is evidence for the former (*Natan et al., 2018*), interfaces involved in nascent chain assembly have not been systematically studied before. Therefore, we cannot exclude the possibility that they have structural features that make them more susceptible to a cotranslational route.

Hydrophobic surfaces play a key role in nucleation theory (*Hermann, 1972*; *Tanford, 1978*; *Chandler, 2005*) and protein folding (*Privalov and Khechinashvili, 1974*; *Chothia, 1975*; *Gething and Sambrook, 1992*), but more importantly, hydrophobicity remains the founding principle of protein–protein recognition theory (*Kauzmann, 1959*; *Chothia and Janin, 1975*). Defined as the buried surface area between subunits, interface size shows correspondence to hydrophobic area because larger interfaces contain more interface core residues (*Levy, 2010*). Conveniently, interface area is relatively simple to compute from structural data (*Hubbard, 1993*; *Kleinjung and Fraternali, 2005*; *Winn et al., 2011*; *Mitternacht, 2016*). Whilst the relationship between interface area and in vitro measured affinity is non-linear (*Eisenberg and McLachlan, 1986*; *Horton and Lewis, 1992*; *Brooijmans et al., 2002*; *Vangone and Bonvin, 2015*), interface area shows remarkable correspondence with subunit dissociation energy, and is reflective of the evolutionary history of subunits within complexes (*Levy et al., 2008*).

We hypothesised that cotranslational interactions may be distinguished from others based upon the areas of the interfaces involved. The size hierarchy of interfaces in protein complexes can be used to predict the order in which their subunits assemble, in good agreement with experimental data (*Levy*

*et al., 2008*; *Marsh et al., 2013*; *Wells et al., 2016*). According to this theory, the largest interfaces in a complex correspond to the earliest forming subcomplexes within the assembly pathway, irrespective of the binding mode. While specific contacts that increase affinity would introduce compositional biases into the sequence space, exerting undue selection pressure on proteomes, variability in interface size can emerge from non-adaptive processes as the organising principle of cotranslational assembly (*Conant, 2009*; *Ahnert et al., 2010*; *Gray et al., 2010*; *Lynch, 2013*; *Leonard and Ahnert, 2019*; *Hochberg et al., 2020*).

In the present study, we address this idea by analysing experimental data on cotranslationally assembling human proteins (*Bertolini et al., 2021*). Our results establish a strong correspondence between cotranslational assembly and subunit interface size. To test whether large interfaces could represent an evolutionary adaptation to cotranslational assembly, we took advantage of the many protein complex subunits that have more than one interface. We compared the areas of first and last translated interfaces in bacterial, yeast, and human heteromeric subunits, and found a clear tendency for the first interface to be larger than the last interface across all species. These findings suggest that large protein complex interfaces have evolved to promote cotranslational assembly.

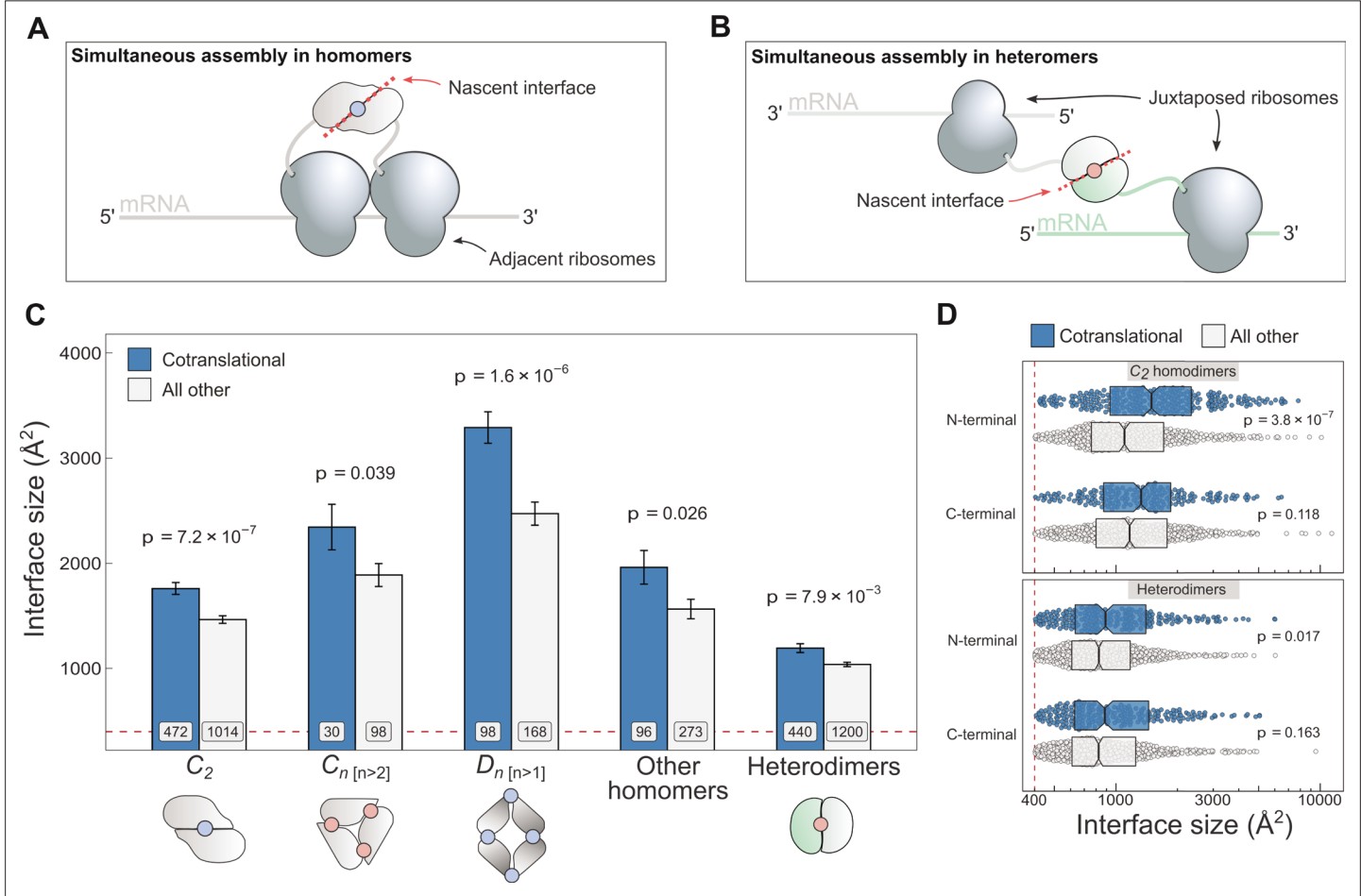

**Figure 1.** Cotranslationally assembling subunits are characterised by large interfaces. (**A**) Schematic representation of (*cis*) simultaneous cotranslational assembly in homomers. (**B**) Schematic representation of (*trans*) simultaneous cotranslational assembly in heteromers. (**C**) Interface size differences between cotranslationally assembling and all other subunits of homomeric symmetry groups and heterodimers. Error bars represent standard error of the mean (SEM) and labels on bars show the number of proteins in each group. The p values were calculated with two-sided Wilcoxon rank-sum tests. Pictograms show the basic structure of symmetry group members, with the blue dots representing isologous and red dots representing heterologous and heteromeric interfaces. (**D**) Interface size distributions of cotranslationally assembling and all other subunits of $C_2$ homodimers and heterodimers, subset by the terminal location of the interface. The p values were calculated with two-sided Wilcoxon rank-sum tests.

The online version of this article includes the following figure supplement(s) for figure 1:

**Figure supplement 1.** Controlling for potential confounders of the cotranslational assembly data.

# Results

## Cotranslationally assembling subunits are characterised by large interfaces

In a recent study, a novel ribosome profiling method was used to identify over 4000 cotranslationally assembling human proteins (*Bertolini et al., 2021*). By design, the method can identify subunits that undergo cotranslational assembly when both subunits are in the process of translation. As recently proposed (*Kamenova et al., 2019*), we refer to this mode of binding as 'simultaneous' assembly (illustrated in *Figure 1A, B*).

To investigate if interface area correlates with simultaneous assembly, we computed the buried surface areas of homomeric and heterodimeric subunits, and subset the results by whether or not the protein was detected to cotranslationally assemble (*Figure 1C*). The arrangement of homomeric subunits with respect to one or more rotational axes allows their classification into symmetry groups. The three most common groups are the twofold symmetric (Schönflies notation, $C_2$), cyclic ($C_{n\,[n>2]}$), and dihedral ($D_{n\,[n>1]}$) complexes, which all have distinct structural and functional characteristics (*Goodsell and Olson, 2000*; *Levy and Teichmann, 2013*; *Bergendahl and Marsh, 2017*) and should therefore be considered separately. For example, members of the cyclic and dihedral symmetry tend to have larger buried surfaces, because they interface with more than one subunit, while $C_2$ symmetric homodimers tend to have larger buried surfaces than heterodimers (*Jones and Thornton, 1996*).

$C_2$ homodimers represent the most highly populated symmetry group and their single isologous interface (i.e. symmetric or head-to-head) makes the analysis simple to perform. In line with our expectation, $C_2$ symmetric subunits that assemble during translation expose 20% larger areas than those that do not (*Figure 1C*; p = 7.2 × 10⁻⁷, Wilcoxon rank-sum test). Considering that the cotranslational assembly annotations are derived from a laboratory technique that uses extensive biochemical fractionation, it is important consider the possibility that larger interfaces would be more persistent to these procedures. We therefore controlled for the potential confounding effect of larger interfaces by setting incrementally higher interface area cutoffs (*Figure 1—figure supplement 1A*), to which the trend appears robust. Note that we have not tested the effects of similar interface cutoffs for the other symmetry groups, due to their much smaller data set size.

Higher-order cyclic complexes are centred on one rotational axis so that every subunit has two distinct interfaces, each with an adjacent protomer. Both interfaces are heterologous (i.e. asymmetric or head-to-tail) and approximately the same size. Cyclic symmetry is potentially confounded by its tendency to form ring-like structures (*Forrest, 2015*), which are ubiquitous components of biological membranes. As a result, membrane-bound complexes are enriched in non-polar amino acids that form the interface with the alkane core of the lipid bilayer. We focussed on the analysis of cyclic homomers that do not localise to the plasma membrane, owing to competing hydrophobic forces exerted by protein–lipid interactions. Despite the limited number of structures available, we detect a significant difference in interface area among soluble members of the cyclic symmetry group (*Figure 1C*), with the mean of cotranslationally forming subunits being 24% larger (p = 0.039, Wilcoxon rank-sum test). Notably, we did not observe the trend in plasma membrane localised cyclic complexes (*Figure 1—figure supplement 1B*).

Dihedral symmetry can be thought of as the stacking of a dimeric or cyclic complex through the acquisition of a twofold axis. All dihedral complexes have isologous interfaces, and those with at least six subunits can have both isologous and heterologous interfaces (e.g. $D_3$ dimers of cyclic trimers). We find that cotranslationally assembling dihedral complexes have on average 33% larger interfaces than those assumed to assemble after their complete synthesis (*Figure 1C*; p = 1.6 × 10⁻⁶, Wilcoxon rank-sum test). A dihedral complex is likely to have evolved from a $C_2$ homodimer if its largest interface is isologous and, conversely, when its heterologous interface is largest, the complex probably arose via a cyclic intermediate (*Levy et al., 2008*; *Marsh and Teichmann, 2014*). When dihedral complexes are grouped by their likely evolutionary history, the trend is present in both groups (*Figure 1—figure supplement 1B*), consistent with that observed in $C_2$ homodimers and higher-order soluble cyclic complexes.

We pooled all remaining homomers, including those with helical and cubic symmetry, and those that are asymmetric, into a single 'other' category, due to their relatively low representation in the human proteome. Altogether, cotranslationally assembling subunits in this heterogeneous category present 25% larger interface areas than other members (*Figure 1C*; p = 0.026, Wilcoxon rank-sum

test). Thus, the interface size trend in cotranslationally assembling complexes appears to hold up across all types of homomers, with the exception of membrane-bound cyclic complexes.

Because simultaneous assembly in heteromers requires two different transcripts positioned in trans (*Figure 1B*), we were curious if they, too, showed a correspondence between cotranslational assembly and interface size. Due to the diverse quaternary structures and assembly pathways associated with heteromeric complexes (*Ahnert et al., 2015*), we focussed on the simplest cases, the heterodimers, which form a single heteromeric interface by the physical interaction of two different proteins. When compared, heterodimers that simultaneously assemble reveal a 15% larger interface area on average than those not detected to cotranslationally assemble (*Figure 1C*; p = 7.9 × 10$^{-3}$, Wilcoxon rank-sum test). Similar to $C_2$ homodimers, the trend in heterodimers is also robust to incremental interface area cutoffs (*Figure 1—figure supplement 1A*), making it unlikely to be an experimental artefact.

The weaker effect size in heterodimers relative to $C_2$ homodimers may be explained by the combination of two factors. First, previous experimental evidence suggests that, in contrast to the simultaneous assembly probed here, heteromers commonly employ the 'sequential' mode of assembly, whereby a subunit in the process of translation recruits a fully synthesised and folded subunit (*Halbach et al., 2009*; *Kassem et al., 2017*; *Shiber et al., 2018*; *Kamenova et al., 2019*; *Panasenko et al., 2019*; *Lautier et al., 2021*; *Seidel et al., 2022*). This mode of assembly has not yet been experimentally probed on a proteome-wide scale, and it is possible that many heterodimers lacking cotranslational assembly annotations in our data set employ sequential assembly. As a result, assuming that interface size plays a role in sequential assembly as well, these unannotated proteins weaken the effect we can detect. Second, it is plausible that another biological process, yet uncharacterised in detail, is responsible for the colocalisation of transcripts and the subsequent subunit assembly (*Wang et al., 2020*; *Chen and Mayr, 2022*), which could make assembly in heteromers less reliant on interface area.

We considered three potentially confounding variables of the ribosome profiling method, which was used to detect cotranslationally assembling proteins. The first is protein length, in part because long polypeptide chains take more time to translate, making it more likely for a cotranslational interaction to come about, and partly because bigger proteins tend to form larger interfaces. Long proteins are also encoded by long transcripts on which structures called di-ribosomes (two ribosomes connected by interacting nascent chains) may persist for a longer time period, potentially leading to their survivorship bias to the observer. In *Figure 1—figure supplement 1C*, we present an analysis where both $C_2$ homodimers and heterodimers are binned by their length into bins containing equal number of structures, and subset by cotranslational assembly. With the exception of long heterodimers, all bins follow the expected interface size trend. More importantly, for both types of complexes, the middle bin, which contains approximately 350–720 residue long proteins and thus covers a large fraction of the human proteome, shows the strongest effect size.

The second variable we accounted for is the confidence-based classification of the cotranslational assembly data set. *Bertolini et al., 2021* employed an elaborate strategy to assign high or low confidence to the protein candidates (detailed in *Bertolini et al., 2021*). However, these high confidence proteins only make up a fifth of all annotations, which prohibits their exclusive use in our analyses. To address this, we leveraged homology models from the SWISS-MODEL repository (*Bienert et al., 2017*; *Waterhouse et al., 2018*) to increase the number of available structures for analysis. With this supplemented structural data set, we found the difference between high confidence and all other subunits (excluding low confidence) to be statistically significant at symmetry level for all homomers (*Figure 1—figure supplement 1D*), but not for heterodimers, consistent with the weaker effect size for heterodimers observed earlier. There are no significant differences observed between high and low confidence proteins for any of the groups. Although we might expect that high confidence proteins should, on average, have larger interfaces than low confidence proteins (assuming that a greater fraction of them represent true cases of cotranslational assembly), we note that the size of the high confidence set is relatively small, especially when split by symmetry group, and that the average interface size of the high confidence proteins is larger for all homomers, except in the very small dihedral group. Overall, we think that the small differences between high and low confidence sets, as well as the small size of the high confidence set, justify our use of the combined sets throughout this study.

The third potential confounder is the location of the interface relative to protein termini. Interactions via N-terminal interfaces are translated earlier, therefore increasing the time available for them to assemble cotranslationally. Given that cotranslationally forming interfaces identified by ribosome

profiling are known to be significantly enriched towards the N-terminus of proteins (*Bertolini et al., 2021*), but that overall, homomeric interfaces tend to be enriched towards the C-terminus (*Natan et al., 2018*), we wished to control for interface location. We classified all interfaces as occurring on either the N- or C-terminal halves of proteins, based on the position of the interface midpoint, which is the residue at which half of the buried surface area of an interface is reached. This comparison is presented for $C_2$ homodimers and heterodimers in *Figure 1D*. In all groups, there is a clear interface size trend wherein cotranslationally assembling subunits have a larger area. More interestingly, however, the trend is only significant and much larger in effect between N-terminally localised interfaces. In fact, N-terminal interfaces are significantly larger than C-terminal interfaces in cotranslationally assembling homodimers (p = 0.021, Wilcoxon rank-sum test). One possible explanation for this is that N-terminally localised interfaces are far more likely to represent cases of genuine cotranslational assembly.

## Interface area is more important than other interfacial contact-based properties for explaining cotranslational assembly

To rule out that interface size is masking a more important property of cotranslationally assembling subunits, we explored other interface features using the same set of $C_2$ symmetric homodimers (*n* = 1486) and heterodimers (*n* = 1640) as shown in *Figure 1A*, which are abundant in the structural data and possess only a single interface, making the results simple to interpret. Because hydrophobicity is essential to protein–protein interactions (*Chothia and Janin, 1975*), we calculated the apolar interface area and compared it to the total area (*Figure 2A, B*). We found that, while the difference in apolar interface area shows a stronger effect among homodimers than the total size (Wilcoxon effect size 0.24, p = 1.8 × 10⁻⁷, vs. 0.228, p = 7.2 × 10⁻⁷), the opposite is observed among heterodimers (Wilcoxon effect size 0.054, p = 0.262, vs. 0.127, p = 7.9 × 10⁻³). The origin of this sharp contrast is likely the fact that heteromeric interfaces are less hydrophobic (*Jones and Thornton, 1996*), and thus complexation is less likely to be primarily driven by the size of the hydrophobic patch of the interface. We also looked at the absolute number of residue–residue contacts within a 5.5 Å radius (*Vangone and Bonvin, 2015*), which echoes the results we obtained with total interface size, but with weaker effects (*Figure 2C*; Wilcoxon effect size, 0.211 for $C_2$, p = 4.4 × 10⁻⁶, and 0.083 for heterodimers, p = 0.08). Next, we employed a contact-based model to estimate binding affinity from the number and character of residue–residue contacts (*Vangone and Bonvin, 2015*). As expected, this analysis revealed that cotranslationally assembling subunits have higher predicted affinities, or lower $\Delta G$ of binding (*Figure 2D*), among both homo- and heterodimers (Wilcoxon effect size, 0.166 for $C_2$, p = 3.1 × 10⁻⁴, and 0.122 for heterodimers, p = 0.011), although these differences are also weaker than those observed with interface size.

We further dissected interfacial contacts based upon chemical character (*Figure 2E*) and interaction type of the residues (*Figure 2F*). The differences within these categories are again weaker than with interface size, the only exception being the contribution of salt bridges to the cotranslational assembly of heterodimers (Dunn's test effect size 0.173, p = 7.7 × 10⁻⁵), which is similarly reflected in the differences between the fraction of charged–apolar and charged–charged contacts. By contrast, the only non-negligible contribution to binding in the cotranslational assembly of homodimers is that of pi–pi interactions (Dunn's test effect size 0.115, p = 7.1 × 10⁻³). While these observations are interesting, they are broadly explained by the apolar interface areas presented in *Figure 2B*. On the one hand, this trend suggests that homodimeric interfaces are made up of more hydrophobic residues, which in turn leave less space for other types of residues, for example charged amino acids that may form salt bridges. This compositional bias in isologous interfaces can, by exclusion, naturally increase the frequency of pi–pi stacked aromatic side chains, a phenomenon that impacts the mutational and evolutionary landscape of symmetric homomers (*Monod et al., 1965*; *Goodsell and Olson, 2000*; *Ponstingl et al., 2005*). On the other hand, heterodimers, whose interfaces are less hydrophobic, accommodate more polar and charged residues that may give rise to specific interactions, in accord with the higher frequency of hydrogen bonds and salt bridges. This notion is consistent with the weak correlation between interface size and binding affinity in heteromers (*Brooijmans et al., 2002*), because in heteromeric protein–protein recognition, interface complementarity is likely to play a more important role than in homomers. Nevertheless, as demonstrated here, interface size shows tremendous utility in discriminating cotranslationally assembling subunits because it is easily calculable and

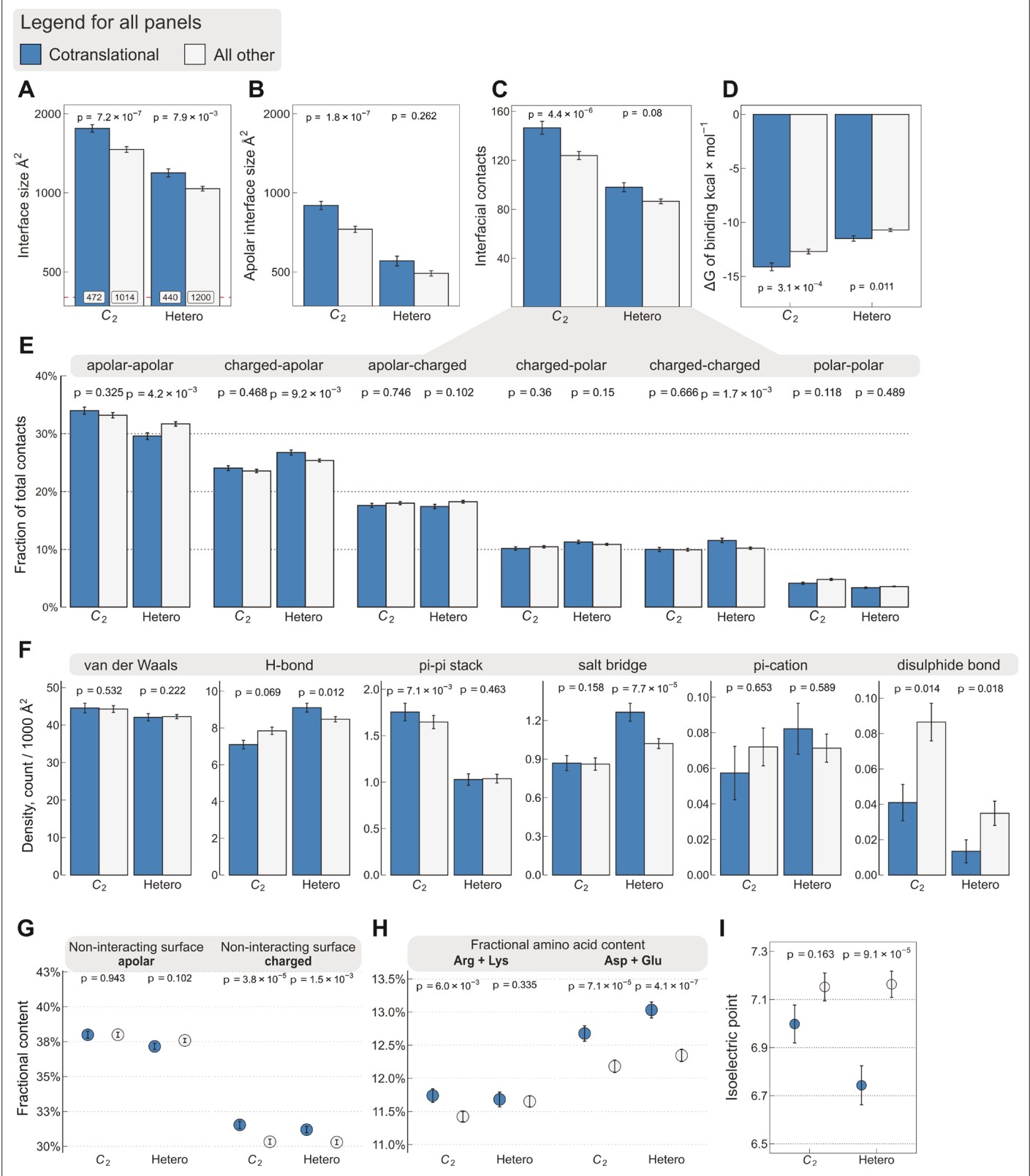

**Figure 2.** Interface area is more important than other interfacial contact-based properties for explaining cotranslational assembly. All panels show the sample mean ± standard error of the mean (SEM) for cotranslationally assembling and all other subunits of human $C_2$ symmetric homodimers and heterodimers. The p values were derived from two-sided Wilcoxon rank-sum tests in panels (**A–D**) and (**I**), and from two-sided Dunn's test of multiple comparisons in panels (**E–H**). Labels on bars in panel (**A**) represent sample sizes. The following parameters are shown. (**A**) Total interface size (Å²). (**B**)

*Figure 2 continued on next page*

Figure 2 continued

Apolar interface size ($Å^2$). (**C**) The absolute number of interfacial contacts. (**D**) Predicted Gibbs free energy ($\Delta G$) of binding (kcal/mol). (**E**) Fraction of residue–residue contacts by chemical character, in descending order of prevalence. (**F**) Specific interaction density (count/1000 $Å^2$), in descending order of prevalence. (**G**) Non-interacting surface apolar ($NIS_a$) and charged ($NIS_c$) residue per cent. (**H**) Fractional content of positively (Arg + Lys) and negatively (Asp + Glu) charged amino acids in the full-length sequence. (**I**) Protein isoelectric point determined with continuum electrostatics on the full monomeric structures.

it is a fundamental property of protein–protein interactions, unlike salt bridges or pi–pi interactions, which do not necessarily occur at every interface.

A linear model developed for the estimation of binding affinity from residue–residue contacts at the interface (*Vangone and Bonvin, 2015*) incorporates coefficients for the terms 'non-interfacial surface apolar/charged' ($NIS_a$ and $NIS_c$, respectively), which are the fraction of apolar or charged surface residues of a subunit in complex, and they have been shown to influence binding affinity (*Kastritis et al., 2014*). We found significant differences in the $NIS_c$ parameter between cotranslationally assembling and all other subunits among both homo- and heterodimers (*Figure 2G*), suggesting that cotranslationally assembling subunits possess a larger proportion of charged residues on the surface than other subunits. To investigate this further, we calculated the fractional content of positively (Arg + Lys) and negatively (Asp + Glu) charged amino acids from protein sequences to see if one charge group in particular is responsible for the trend. This analysis is possible because other than the relatively rare internal charges (*Hendsch and Tidor, 1994*; *Kajander et al., 2000*), most charges tend to be on the surface, an assumption that is supported by our structural data, in which nearly 83% of charged residues have more than 25% relative accessible surface area in the monomeric protein (*Levy, 2010*). We found that negatively charged amino acids in cotranslationally assembling subunits are overrepresented relative to other subunits (*Figure 2H*; Dunn's test of multiple comparisons, $C_2$, p = 7.1 × $10^{-5}$ and heterodimers, p = 4.1 × $10^{-7}$), and we also detect a small but significant enrichment in positively charged amino acids, but only in homodimers (p = 6.0 × $10^{-3}$). We further support these observations with isoelectric points calculated in continuum electrostatics at physiological pH and salt concentration from the human AlphaFold structures (*Tunyasuvunakool et al., 2021*), which represent the full-length monomers translated on ribosomes. This analysis revealed that the net enrichment in charged amino acids on the surfaces of cotranslationally assembling heterodimeric subunits results in a lower isoelectric point than in other subunits (*Figure 2I*; Wilcoxon rank-sum test, p = 9.1 × $10^{-5}$).

What may explain the finding that cotranslationally assembling subunits display more negative surface charges than other proteins? We believe there are four mutually non-exclusive hypotheses that are compatible with the observation. The first is based on the work of *Kastritis et al., 2014*, who proposed based on alanine scanning mutagenesis experiments that polar and charged residues of non-interacting surfaces contribute to binding affinity. Second, one might argue that the role of charged residues on the surface is to counteract the strong water-orientation forces exerted at large interfaces by supporting protein solubility through favourable interactions with water molecules (*Kramer et al., 2012*) and ions (*Linse et al., 1988*). The third idea concerns the ribosome:nascent chain interaction, where negative charges could help avoid unproductive interactions with the ribosome surface (*Cassaignau et al., 2021*; *Deckert et al., 2021*), thus facilitating cotranslational folding and assembly. The fourth scenario would be attributable to a proteome-wide effect, whereby the higher the abundance of a protein, the more its surface has been shaped by evolution for optimal 'stickiness' to combat non-specific interactions upon molecular crowding (*Levy et al., 2012*). While further analysis of this effect is out of the scope of this study, using pooled homo- and heterodimer data, we detect a weak but significant Spearman correlation of 0.18 (p = 1.7 × $10^{-23}$) between the $NIS_c$ parameter and HEK293-specific active ribosome count (an abundance measure, *Clamer et al., 2018*), which corroborates the fourth hypothesis.

## Larger and earlier-assembling interfaces tend to form cotranslationally in heteromeric subunits with multiple interfaces

Having confirmed that subunit interface size correlates with cotranslational assembly, we next wanted to see if this trend applies within single subunits that have more than one interface. In other words, do multi-interface heteromeric subunits also employ their largest interface during the course of simultaneous assembly? A multi-interface heteromeric subunit forms at least two distinct interfaces with two

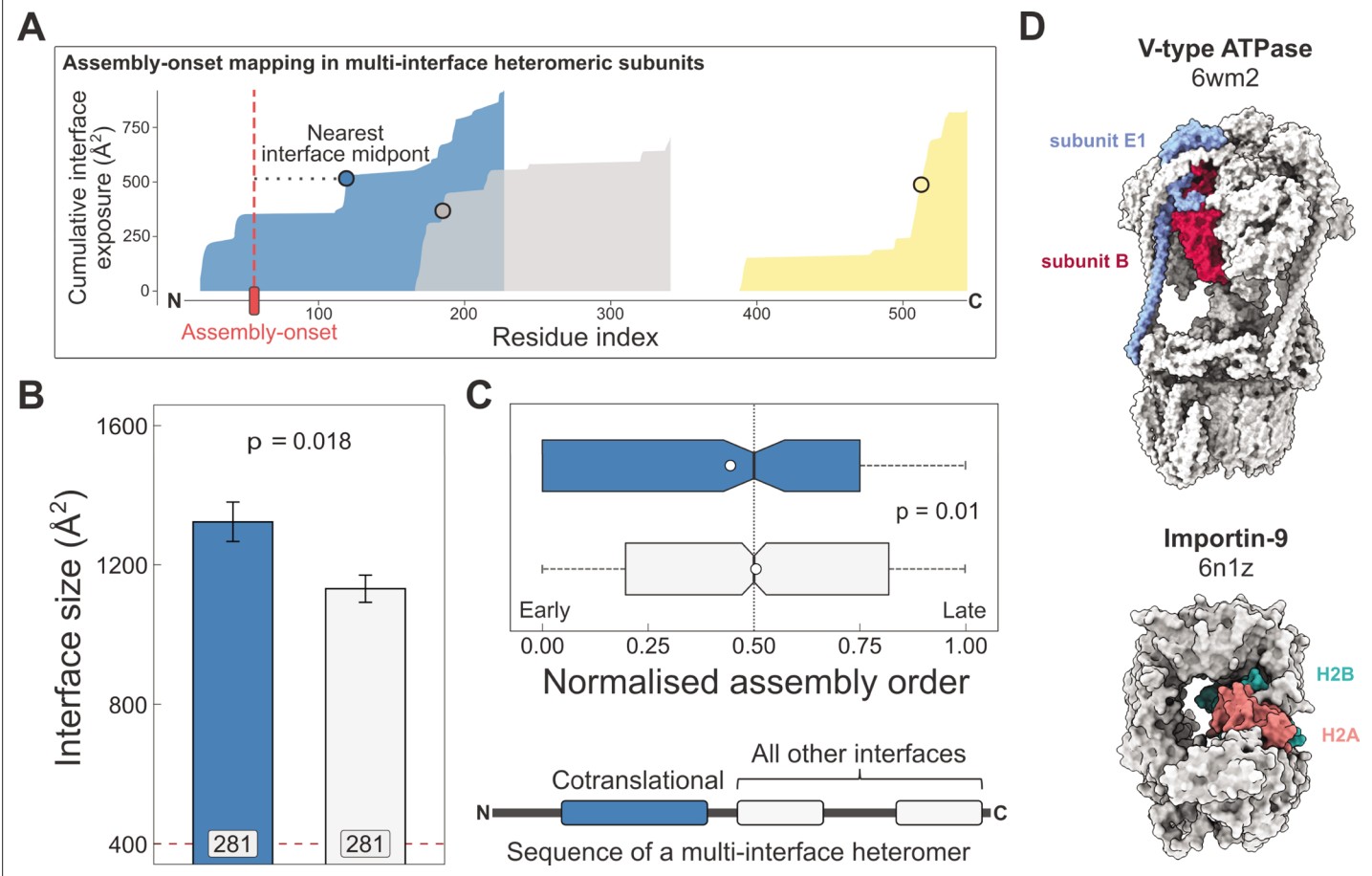

**Figure 3.** Larger and earlier-assembling interfaces tend to form cotranslationally in heteromeric subunits with multiple interfaces. (**A**) Visual representation of the interface mapping protocol. The area plot shows the cumulative interface area build-up of individual interfaces during translation, which are shown in different colours. The midpoints (dots) are residues at which half of the eventually formed area is exposed. Assembly-onsets determined by *Bertolini et al., 2021* are mapped to the nearest midpoint on condition that it is not a homomeric interface. (**B**) Pairwise comparison of cotranslationally forming (in the simultaneous mode) interfaces of multi-interface heteromeric subunits to the mean of all other heteromeric interfaces on them. For visual aid, see line diagram under panel (**C**) Error bars represent standard error of the mean (SEM) and labels on bars show the number of proteins in each group. The p value was calculated with the Wilcoxon signed-rank test. (**C**) Pairwise comparison of the normalised assembly order in 201 complexes between cotranslationally forming and all other heteromeric interfaces. The normalised assembly order is a 0-to-1 scale where 0 and 1 represent the first and the last steps of the predicted assembly pathway. The p value was calculated with the Wilcoxon signed-rank test. (**D**) Two examples of simultaneous cotranslational assembly between subunit pairs in heteromeric complexes: the subunits E and B1 of the V-type ATPase (pdb: 6wm2), and importin-9 with histone H2A (6n1z).

other proteins in a complex that contains at least three subunits. Because of the interface hierarchy that exists within protein complexes (*Levy et al., 2008*; *Marsh et al., 2013*), we hypothesised that the largest interface, which is most likely to assemble earliest, should also be more likely to cotranslationally assemble.

To perform an analysis at the multi-interface level, we made use of the assembly-onset positions determined for every protein in the cotranslational assembly data set (*Bertolini et al., 2021*). An assembly-onset is a single residue in the protein sequence, whose codon is being decoded by the ribosome at the time of cotranslational assembly. In order to identify which interface the assembly-onsets belongs to, we mapped them to the closest interface midpoint in the linear protein sequence, as illustrated in *Figure 3A*. This is to avoid biases from large interfaces, which have many more interface residues and therefore a higher probability that an assembly-onset would map to them if the interface was not compressed into a single midpoint residue.

Using this method, we identified 281 interfaces on multi-interface heteromeric subunits that may form in a simultaneous cotranslational fashion. To see if these correspond to the largest interfaces

within each protein, we calculated the mean interface area for all other interfaces on these subunits to be able to perform a paired statistical test. Our results show that the identified cotranslationally assembling interfaces are indeed larger by 19% than other interfaces on these subunits (*Figure 3B*; p = 0.018, Wilcoxon signed-rank test).

We wished to put these interfaces into the context of their full complexes. Do simultaneously forming interfaces represent early forming subcomplexes that then initiate further assembly events, since the first step of a protein complex assembly pathway is the most likely to occur cotranslationally (*Wells et al., 2015*)? Although the largest interface in a complex is always predicted to assemble earliest in the assembly pathway, subsequent steps are non-trivial because they can involve multiple subunit:subunit interfaces (*Ahnert et al., 2015*; *Levy et al., 2008*; *Marsh et al., 2013*). To answer this question, we predicted the assembly steps of heteromeric complexes on the basis of their structures (*Wells et al., 2016*). This analysis revealed that the identified interfaces tend to form much earlier than other heteromeric interfaces in the complexes (*Figure 3C*; p = 0.01, Wilcoxon signed-rank test). Another interpretation of this can be given by classifying assembly steps into 'early' and 'late', depending on their normalised assembly order (*McShane et al., 2016*), which is a 0-to-1 scale indicating the first-to-last steps of a pathway, where we defined early steps with values less than or equal to 0.5. According to this, a simultaneously forming interface is 1.7 times more likely to form early (180 [67%] of 270 vs. 633 [54%] of 1171; p = $9.5 \times 10^{-5}$, Fisher's exact test).

Some of the identified interfaces belong to complexes that have been shown to use cotranslational assembly routes, such as the proteasome (*Panasenko et al., 2019*) and subunits of the transcription initiation complex (*Kamenova et al., 2019*). However, many are not yet described in the literature, for example, the loading of histone H2A onto importin-9 (*Figure 3D*), which has been reported to act as a storage chaperone while transporting a histone dimer to the nucleus (*Padavannil et al., 2019*). Another example is the V-type ATPase (*Figure 3D*), whose catalytic A and B subunits have been tested for their ability to assemble in the sequential mode with a negative result (*Shiber et al., 2018*), but our structural approach using the assembly-onset identified the E1 subunit to form in the simultaneous mode with the catalytic B subunit. Although these two subunits can undergo major structural rearrangements in the complex, the same B subunits stay in contact with the same E1 subunits across all the observed conformational states (*Vasanthakumar et al., 2022*). In fact, such large post-translational conformational rearrangements may be common in cotranslationally forming interfaces, given that proteins with larger interfaces will have an inherent tendency to be more flexible (*Marsh and Teichmann, 2014*).

## Evolutionarily more ancient subunits of complexes are more likely to undergo cotranslational assembly

Protein complexes are under evolutionary selection to minimise misassembly (*Marsh et al., 2013*; *Leonard and Ahnert, 2019*), meaning that over evolutionary timescales, ordered subunit assembly has been prioritised in cells (*Wells et al., 2016*). These findings have led to the formulation of the interface size hypothesis, which posits that the assembly pathway of a protein complex parallels with its evolutionary history. A simple proxy for predicting the steps of such pathways is interface size, which demonstrates exceptional correspondence with in vitro complex assembly–disassembly analyses (*Levy et al., 2008*; *Marsh et al., 2013*).

Naturally related to the interface size hypothesis is another trend that reflects the time it takes for a newly emerged interface to become strengthened by evolution, either beacause of a functional association between the proteins, or via entrenchment in a neutral ratchet (*Kim et al., 2006*; *Dayhoff et al., 2010*; *Hochberg et al., 2020*). To address this, we grouped heteromeric protein complex subunits based on the phylogenetic class in which a protein's encoding gene first appeared, that is protein age (*Liebeskind et al., 2016*). In addition to the human proteins, we also employed a data set of yeast complexes. Due to the smaller size of the yeast data set, we supplemented it with heteromeric models computed for yeast core complexes, which were determined using residue coevolution inferred from paired multiple sequence alignments, followed by deep learning-based structure prediction of subunit pairs with experimental evidence to interact (*Humphreys et al., 2021*). In *Figure 4A*, we show that the average interface size of the groups, for both human and yeast, is ordered almost perfectly according to evolutionary age of both homo- and heteromers. When we ask what percentage of each protein age group was found to assemble cotranslationally by *Bertolini*

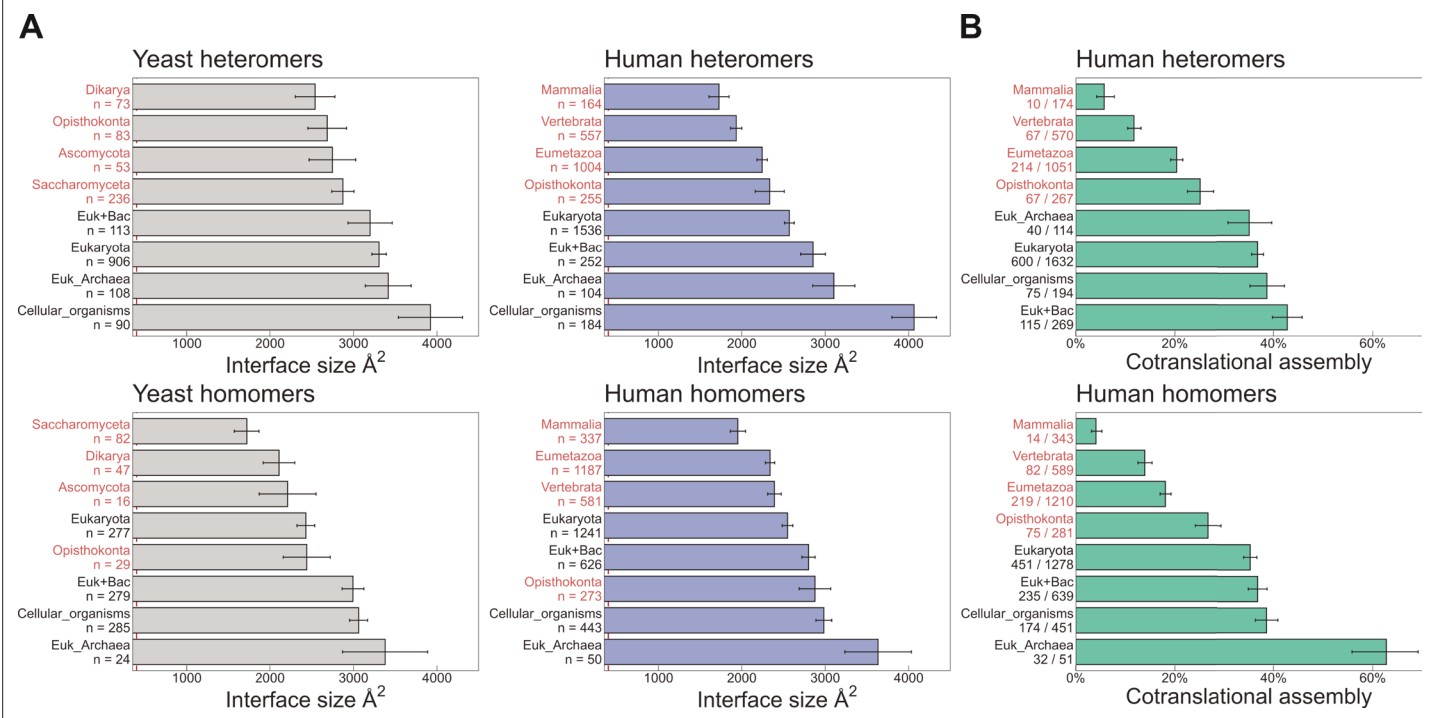

**Figure 4.** Evolutionarily more ancient subunits of complexes are more likely to undergo cotranslational assembly. (**A**) Average (mean ± standard error of the mean [SEM]) interface sizes of yeast and human homo- and heteromeric subunits grouped by the evolutionary age of the protein. Age group labels coloured in red are defined as 'more recent' proteins, while those in black represent 'ancient' proteins. Numbers under labels represent the number of distinct proteins in the given age group. Homomeric interface sizes are a pool of experimentally determined structures and SWISS-MODEL homology models. (**B**) The frequency (%) of cotranslational assembly, as detected by *Bertolini et al., 2021*, in the different protein age groups, split into homo- and heteromers. Heteromer annotations were supplemented with those contained in hu.MAP2.0 (*Drew et al., 2021*). Fractions under labels denote the number of cotranslationally assembling proteins out of the total in the given age group. Error bars represent 68% Jeffrey's binomial confidence intervals.

*et al., 2021*, there is a remarkable agreement between older proteins with larger interfaces having a higher frequency of cotranslational assembly (*Figure 4B*). Older subunits are also more likely to be found in more than one complex than younger subunits (*Saeed and Deane, 2006*; *Drew et al., 2021*); therefore cotranslationally assembling proteins are expected to be enriched in moonlighters. Indeed, it has recently been shown among components of the nuclear pore complex that some subunits preferentially assemble cotranslationally with one partner, but not with a different partner from another complex (*Seidel et al., 2022*).

## N-terminal interfaces tend to be larger than C-terminal interfaces supporting evolutionary selection for cotranslational assembly

There are two possible explanations for the observation that cotranslationally forming interfaces tend to be larger. First, larger interfaces may be inherently more likely to form cotranslationally because their assembly is more energetically favourable. In this scenario, cotranslational assembly has not been evolutionarily selected for; instead, the larger interfaces are simply more likely to be formed while the protein is still in the process of being translated, without providing any functional benefit. Alternatively, cotranslational assembly may have been selected for, for example, because it increases the efficiency of assembly and avoids potentially damaging non-specific interactions. Here, large interfaces have evolved to increase the level of functionally beneficial cotranslational assembly.

One way to distinguish between these two scenarios is to compare the sizes of N- and C-terminal interfaces. Regardless of whether cotranslational assembly occurs simultaneously (*Figure 1B*) or sequentially (*Figure 5A*), due to vectorial synthesis on the ribosome, N-terminal regions of proteins are more likely to be involved in binding events during translation. Therefore, if cotranslational assembly is adaptive, we would expect that N-terminal interfaces in multi-interface heteromeric subunits, which

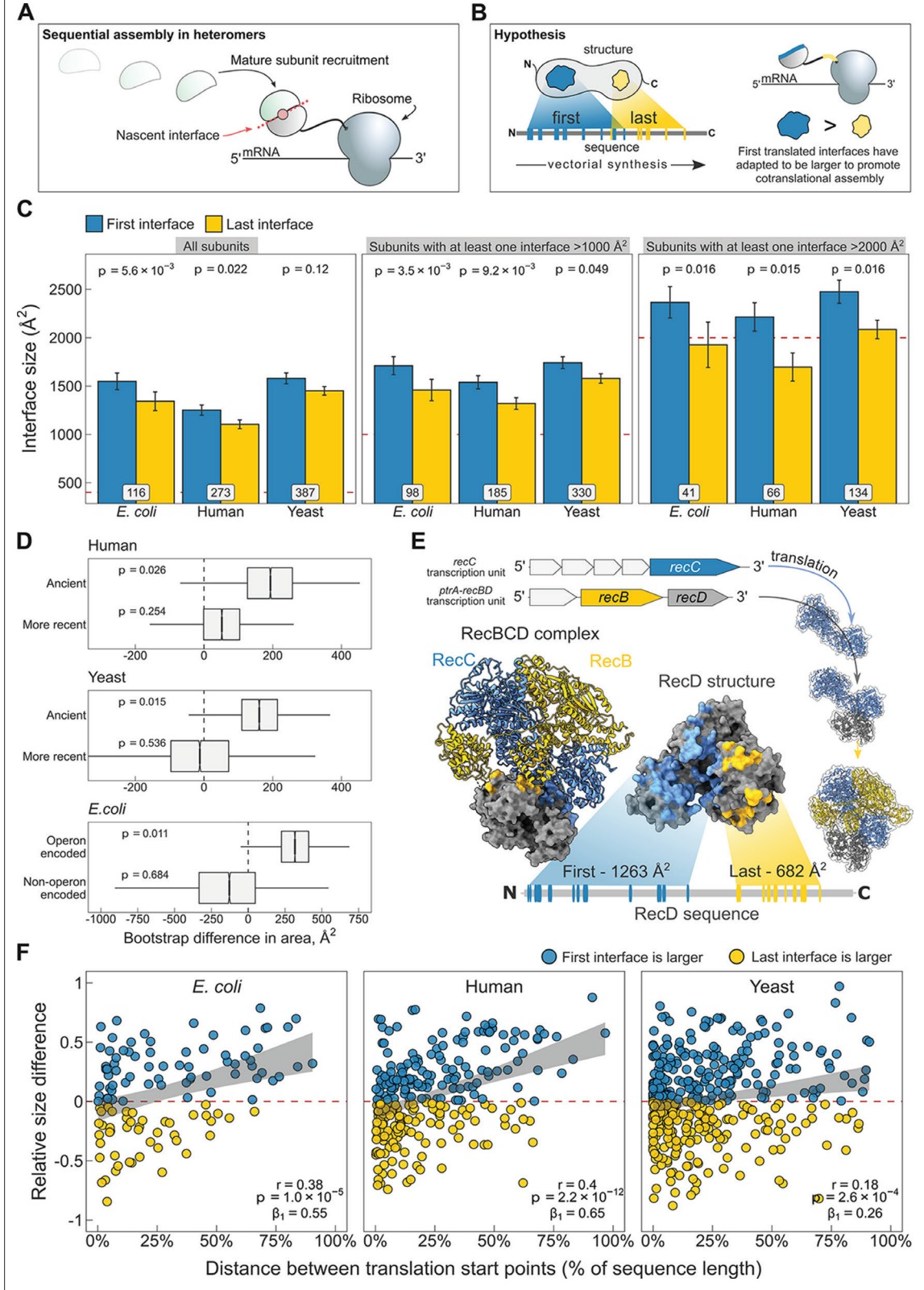

**Figure 5.** N-terminal interfaces tend to be larger than C-terminal interfaces supporting evolutionary selection for cotranslational assembly. (**A**) Schematic representation of sequential cotranslational assembly in homomers. (**B**) Diagrammatic representation of the hypothesis test of the adaptive model of cotranslational assembly. (**C**) Area differences between the first and last translated interfaces in multi-interface heteromeric subunits across the species. Panels are ordered by the area cutoffs, 400, 1000, and 2000 Å², which are satisfied if either the first or the last interface is larger than the

*Figure 5 continued on next page*

*Figure 5 continued*

given cutoff. Error bars represent standard error of the mean (SEM) and labels on bars show the number of proteins in each group. The p values were calculated with Wilcoxon signed-rank tests. (**D**) Bootstrap distributions of the area difference between the first and the last translated interfaces within two different categories. The first category (top two boxplots) is protein age, whereby yeast and human multi-interface heteromeric subunits are classified into 'ancient' and 'more recent' groups. In the second category (bottom boxplot), bacterial subunits are grouped based on whether or not they are encoded in operons. Positive values represent a larger first translated interface. The p values were calculated from $10^4$ bootstrap resamples with correction for finite testing. (**E**) Example of an operon-encoded complex, the RecBCD nuclease (pdb: 5ld2). In the linear sequence of RecD, the interface with RecC is translated first, and that with RecB is last. The RecD:RecC interface is twice the area of the RecD:RecB interface, likely to promote cotranslational subunit recruitment. (**F**) Correlation between the relative distance of translational start points and the relative area difference of the first and last translated interfaces. Shaded lines represent the 95% confidence interval of the regression line. The Pearson's correlation coefficient r, its p value, and the regression coefficient $\beta_1$ are shown in the panels.

The online version of this article includes the following figure supplement(s) for figure 5:

**Figure supplement 1.** Additional analyses supporting the results shown in *Figure 5*.

---

will be translated first, should show a significant tendency to be larger than C-terminal interfaces, as illustrated in *Figure 5B*.

To address the question, we selected experimentally determined heteromeric complex structures from three model proteomes: *Escherichia coli*, *Saccharomyces cerevisiae* (yeast), and *Homo sapiens* (human). For each heteromeric subunit, we defined the first interface as the one that exposes the most N-terminal interface residue in the linear protein sequence, and, to treat the termini symmetrically, the last interface was defined as the one that exposes the last interface residue, that is the first interface residue from the C-terminal direction.

When we compare the areas of the first and last translated interfaces in heteromeric subunits across species, we find the first interface to be larger (*Figure 5C*; full distribution in *Figure 5—figure supplement 1A*). The strongest effect is measured in *E. coli*, where the first interface is larger than the last interface in 60% of cases, and on average is 205 Å² (15%) larger (p = 5.6 × 10⁻³, Wilcoxon signed-rank test). In humans, the trend is weaker, with the first interface larger in 52% of cases, being 147 Å² (13%) larger on average (p = 0.022). In yeast, although the first interfaces are 130 Å² (9%) larger on average, the difference is not significant. Averaging over the three species, we observe first interface to be larger than the last interface in 54% of multi-interface subunits.

The above analysis includes all multi-interface heteromeric subunits. However, some of these contain only very small interfaces, and would therefore be unexpected to undergo cotranslational assembly via either interface, thus reducing the observed tendency for N-terminal interfaces to be larger. Indeed, when we filter the subunits to exclude those with only small interfaces, including only those where the size of at least one of the two interfaces is larger than 1000 or 2000 Å², we find that the trend gets much larger across all three species (*Figure 5C*). When using the 2000 Å² filter, we observe that N-terminal interfaces are on average 23%, 31%, and 19% larger for *E. coli*, human, and yeast, respectively, significant for all. Averaging over the three species, we observe first interface to be larger than the last interface in 64% of cases. Thus, when considering multi-interface subunits of heteromeric complexes where at least one interface is >2000 Å², the N-terminal interface is larger than the C-terminal interface nearly two thirds of the time.

Building on our earlier observation that cotranslationally assembling proteins tend to be older in evolutionary terms (*Figure 4B*), we asked if a protein's age could also influence the size of the N-terminal interface, assuming that over time selection would work to minimise misassembly by favouring larger first translated interfaces and ensuring cotranslational assembly between the correct subunit pair. We split the multi-interface subunits from human and yeast into 'ancient' and 'more recent' categories (see Materials and methods). We then generated bootstrap distributions of the area difference between the first and the last translated interfaces in yeast and human multi-interface heteromeric subunits within the respective age categories. The analysis revealed that the difference between the areas of the first and last translated interfaces is significantly larger in ancient proteins in both yeast and human (*Figure 5D*; mean difference of 163 Å², p = 0.015 and 192 Å², p = 0.026, respectively), but not in younger heteromers. We hypothesise that this is because there has not been enough time for selection to considerably act on the assembly of these subunits, or because newer subunits tend to be required less for cotranslational interactions due their later assembly or functional roles they

play. Thus, these analyses support that large, typically ancient, interfaces have evolved to promote cotranslational assembly.

Our attention was next drawn to *E. coli*, which demonstrated the strongest first versus last interface size trend. In prokaryotes, many heteromeric complexes are encoded by operons, where the different subunits are translated off of the same polycistronic mRNA molecules. Early studies in bacteria indicated that operon gene order is correlated to physical interactions between the encoded proteins (*Mushegian and Koonin, 1996*; *Dandekar et al., 1998*). Further investigation laid down theoretical, mechanistic, and evolutionary evidence in support of this (*Sneppen et al., 2010*; *Shieh et al., 2015*; *Wells et al., 2016*). Cotranslational assembly is likely to be particularly common in operon-encoded heteromers, given that the translation of different subunits is inherently colocalised. We hypothesised that the tendency for N-terminal interfaces to be larger should be stronger in operon-encoded *E. coli* heteromers, compared to those that are not operon encoded.

We illustrate the example of the RecBCD nuclease in *Figure 5E*. Genes of the subunits are located in adjacent loci encoding transcriptional units for RecC and RecB/D. One study reported that purification of RecD is complicated by the formation of inclusion bodies, while the other two subunits remain in the soluble fraction (*Marsh and Teichmann, 2014*). Moreover, a genetic analysis suggested that partially folded RecC and RecD might interact during translation, or that RecC forms a complex with RecB first, onto which RecD is then assembled (*Amundsen et al., 2002*). The regulatory subunit RecD has two interfaces well separated in the sequence, where the interface with RecC is translated first. One might imagine that the nascent chain of RecD forms a complex with mature subunit of RecC, having double the interface area to accommodate RecC than that for RecB. In this scenario, the assembly efficiency is not only maximised by gene order reducing stochasticity, but also by cotranslational assembly minimising the need for post-translational association.

To further test whether large interfaces could have been selected to promote cotranslational assembly, we acquired annotations derived from RNA sequencing data sets (*Chetal and Janga, 2015*) to group heteromers from *E. coli* according to whether or not they are encoded by operons. We again generated bootstrap distributions of the area difference between the first and the last translated interfaces to visualise and derive a probability (*Figure 5D*). In agreement with the above idea, we found that the size difference between the two interfaces is significantly larger in operon-encoded multi-interface heteromeric subunits, favouring the first interface (mean area difference of 317 Å$^2$, p = 0.011).

We speculate that the first versus last interface trend may be the hallmark of sequential cotranslational assembly (*Figure 5A*), rather than that of the simultaneous mode (*Figure 1B*). *Bertolini et al., 2021* have suggested that simultaneous assembly is predominantly employed for the formation of homomeric protein complexes, which could mean sequential assembly is the more common cotranslational assembly mode in heteromers. The strong trend in *E. coli* also supports this idea, because polycistronic gene structure is more compatible with sequential cotranslational assembly (*Shieh et al., 2015*). In eukaryotes, large complexes and subunits of lowly abundant complexes may require an additional biological process to ensure their transcripts are colocalised for simultaneous assembly and

**Table 1.** Table of yeast multi-interface heteromeric subunits, which have been shown to utilise the sequential mode of cotranslational assembly.

The yeast Set1, part of the COMPASS complex, binds multiple partners during its translation process (*Halbach et al., 2009*), but the order of these assembly steps is uncertain. In a partial structure of the histone methyltransferase complex (6b × 3), Set1 is found to have two biologically significant interfaces, the first with Swd3 and the last with Swd1. For the rest of the cases, the mature partners are known and the available structural data support a model where the first translated interface is larger relative to the last translated interface, likely to promote cotranslational subunit recruitment.

| Study | Nascent chain | Mature partner | First (Å$^2$) | Last (Å$^2$) | Source |
|---|---|---|---|---|---|
| *Halbach et al., 2009* | Set1 | uncertain | 1,467 | 1,025 | 6b × 3 |
| *Shiber et al., 2018*; *Fischer et al., 2020* | Fas2 | Fas1 | 4,226 | 484 | 6ql9 |
| *Shiber et al., 2018* | Pfka1 | Pfk2 | 7,796 | 5,749 | 3o8o |
| *Shiber et al., 2018* | Gcn3 | Gcd2 | 1,210 | 687 | *Humphreys et al., 2021* |
| *Panasenko et al., 2019* | Rpt2 | Rpt1 | 4,226 | 484 | 6fvt |

to facilitate further assembly steps (*Keene, 2007*; *Chen and Mayr, 2022*). Sequential assembly, on the other hand, may have evolved to exploit large interface areas for the recruitment of partner subunits. This can be conceptualised as the 'bait and prey strategy' of cotranslational assembly, in which a large nascent interface represents the 'bait' bound by a fully folded 'prey' subunit. Although a proteome-scale data set of cotranslationally assembling proteins is not available for yeast, we have identified case studies of five multi-interface heteromeric subunits in yeast that use the sequential assembly mode, and found that all five subunits follow the first versus last interface size trend (*Table 1*). To substantiate the model further, we removed from the human data set those proteins that were identified by *Bertolini et al., 2021*, that is those that simultaneously assemble. Strikingly, removal of these proteins increases the size difference between the first and the last translated interfaces from 13% to 18% (*Figure 5—figure supplement 1B*; p = 0.014, Wilcoxon signed-rank test). One explanation is that the remaining heteromers are enriched in sequential assembly, and thus exhibit a greater difference.

A property that would be consistent with the above model is interface separation. The later the translation of the last interface starts relative to the first, the higher the chance that assembly of the first interface will be undisturbed, free of competition with the partner subunit of the last interface. Therefore, we hypothesised that the distance between translation start points of the first and last interfaces, which are the earliest emerged interface residues of each, should correlate with the size difference in favour of the first interface. Because of large variances in protein length and interface size, we normalised the translational distance between the first and last interfaces as the percentage of the protein's sequence length, and scaled the area difference by the sum of both interfaces. *Figure 5F* shows the correlation between the separation of translation start points and the area differences of the first and the last interfaces (absolute values shown in *Figure 5—figure supplement 1C*). As expected, increasing the distance between the start points monotonically increases the extent of the area difference across all species. To rule out that the interface separation metric is confounded by sequence length, we split the structures into less and more than 400 amino acids, which is the pan-species mean of sequence lengths. In both subsets, there is a pronounced preference in all species for a larger first interface when the separation is high (*Figure 5—figure supplement 1D*). The causal direction of this effect, whether it reflects that cotranslational assembly happens more often in high degrees of interface separation, or that separation is driven by selection for cotranslational assembly, remains to be addressed.

## Discussion

It has long been understood that interface area is important for assembly, but the capacity in which it shapes the hierarchy of individual interfaces on subunits remained elusive. In this study, we first combined information from ribosome profiling with structural data on protein complexes to probe the importance of interface area in the process of cotranslational assembly. Our results demonstrate a strong correspondence between interface size and cotranslational subunit binding. Inspired by this, we set out to test an important question about the biological significance of cotranslational assembly: do large interfaces give rise to cotranslational assembly because of simple energetic reasons or do they reflect an evolutionary adaptation for a functional benefit? We found a clear trend across three evolutionarily distant species for the first translated interface of heteromeric subunits to be larger, suggesting that large interfaces have evolved to promote cotranslational assembly.

While our results support the adaptive hypothesis of interface size, it is not entirely inconceivable that cotranslational assembly represents a ratchet-like mechanism of constructive neutral evolution (*Gray et al., 2010*; *Hochberg et al., 2020*), whereby a drift in interface properties creates ideal conditions for assembly on the ribosome. Reversion to the post-translational route is prevented by the accumulation of mutations that are neutral in the subunit entrenched in cotranslational assembly, but would otherwise be deleterious in the ancestor. A similar neutral process may sustain the differences in interface area presented in this study.

In light of current knowledge of bacterial operon structure and translation regulation, it is not surprising that we observed the strongest trend among *E. coli* heteromers with respect to the size of the interface that first emerges from the ribosome. Supposedly, the effect is attributable to the widespread sequential assembly between mature subunits and nascent chains, reflecting the mechanism of effective subunit recruitment by large N-terminal 'bait' interfaces. An interesting question for laboratory experiments is whether operon-encoded heteromers can assemble in the simultaneous

mode, providing that the structural organisation of the bacterial polysome allows for such a precise coordination (**Brandt et al., 2009**).

How does a large interface area help translating ribosomes find one another? Its benefit in homomers for facilitating cotranslational assembly is clear, because the subunits are localised to the same mRNA, and large interfaces are more likely to form interactions before translation is complete. In heteromers, one hypothesis argues the involvement of RNA-binding proteins that orchestrate transcript colocalisation (**Keene, 2007**; **Chen and Mayr, 2022**), from where similar rules may apply to heteromer assembly as for homomers. A more parsimonious hypothesis is formed on the observation that cotranslational assembly can result in transcript colocalisation, which is ablated when subunit affinity is decreased (**Heidenreich et al., 2020**). This may suggest that affinity, which correlates strongly though imperfectly with interface size (**Brooijmans et al., 2002**; **Vangone and Bonvin, 2015**), can play a role in the colocalisation of transcripts belonging to the same complex.

Many more topics of inquiry remain open for future studies. Analogous to protein folding, cotranslational assembly can be thought of as a hydrophobic collapse that shapes the quaternary structure of the complex. As with folding in the cell, the involvement of other factors must not be overlooked. A wide array of ribosome-associated chaperones are vital for nascent chain homeostasis (**Oh et al., 2011**; **Döring et al., 2017**; **Koldewey et al., 2017**; **Shiber et al., 2018**) and the degree to which they choreograph assembly steps is yet to be elucidated.

Attention should be paid to the far-reaching genetic consequences of cotranslational assembly (**Natan et al., 2017**). How much does transcript-specific assembly buffer the dominant-negative effect, and what does it mean in the context of human genetic disease (**Bergendahl et al., 2019**)? This effect requires mutant subunits to be stable enough to assemble into complexes, and thus the impact of the mutations tends to be milder at the structural level than of other pathogenic mutations (**McEntagart et al., 2016**), making them exceptionally difficult to detect using the existing variant effect predictors (**Gerasimavicius et al., 2021**). Interestingly, cotranslational assembly should actually make the dominant-negative effect less common in homomers, because it can limit the mixing that occurs between wild-type and mutant subunits. It remains to be seen whether dominant-negative mutations are in fact less common in cotranslationally assembling complexes.

Finally, our results build on evidence from the past decade and emphasise the importance of protein complex assembly at the translatome level. Although evolutionary selection against N-terminal interface contacts to avoid premature assembly was previously found in homomers (**Natan et al., 2018**), here we report an opposite phenomenon in which proteins that do cotranslationally assemble sustain large N-terminal interfaces in order to promote cotranslational subunit recruitment. We expect that our observation will be supported by experimental data once the proteome-wide detection of sequentially assembling heteromeric subunits is made possible.

## Materials and methods
### Protein structural data sets

Starting from the entire set of structures in the Protein Data Bank (**Berman et al., 2000**) on 2021-02-18, we searched for all polypeptide chains longer than 50 residues with greater than 90% sequence identity to *H. sapiens*, *S. cerevisiae*, and *E. coli* canonical protein sequences. When proteins mapped to multiple chains, we selected a single chain sorting by sequence identity, then by the number of unique subunits in the complex, and then by the number of atoms present in the chain. Only biological assemblies (pdb1) were used and symmetry assignments were taken directly from the PDB. Polypeptides formed by cleavage were excluded. In the generation of the multi-interface heteromeric subunit data sets, to exclude proteins with yet uncharacterised interfaces, chains with an at least 70% complete structure were considered and only included if they formed interface pairs >800 Å$^2$ with at least two different subunits. To supplement the smaller yeast data set, computed structures of yeast core complexes were downloaded from the ModelArchive link provided by **Humphreys et al., 2021**. For downstream analysis, mmCIF files were converted into standard Brookhaven PDB format and the chains were mapped to genes using the table provided on ModelArchive. Homology models of yeast and human homomeric complexes were obtained from the SWISS-MODEL repository (version 2022_02) (**Bienert et al., 2017**; **Waterhouse et al., 2018**). When a protein's UniProt accession number mapped to multiple homology models, we selected a single model ranking by the number of subunits

in the complex, followed by the length of the modelled chain. Symmetry groups of the homology models were assigned with the software AnAnaS (*Pagès and Grudinin, 2018*; *Pagès et al., 2018*). In the analysis of N- versus C-terminal interface sizes, we excluded very large heteromeric complexes, defined as those containing ≥10 subunits. This is because of the previous evidence that predicting assembly order based on interface size in very large complexes is not as accurate (*Marsh et al., 2013*; *Ahnert et al., 2015*), likely because of the many intersubunit interfaces these complexes possess.

## Calculation of interface area-related properties

Interface areas of SWISS-MODEL homology models were calculated with FreeSASA (*Mitternacht, 2016*) using the default surface probe radius of 1.4 Å. Residue-level pairwise interfaces in complexes derived from the PDB and from *Humphreys et al., 2021* were calculated between all pairs of subunits using AREAIMOL from the CCP4 suite (*Winn et al., 2011*) with a probe radius of 1.4 Å. The interface was defined as the difference between the solvent accessible surface area of each subunit in isolation and within the context of the full complex. Apolar interface area was calculated from the residue-level data by classifying A, F, G, I, L, V, M, P, and Y amino acids as apolar (*Vangone and Bonvin, 2015*). An area cutoff of >400 Å$^2$ was used for homomeric subunits and individual interfaces of multi-interface heteromeric subunits derived from the PDB to exclude potential crystallographic interfaces and to restrict the analyses to biologically significant interfaces. Assembly order was computed by predicting the assembly pathway assuming additivity of pairwise interfaces in each complex (*Marsh et al., 2013*), and implemented with the *assembly-prediction* Perl package (*Wells et al., 2016*).

The relative interface location was calculated according to the formula:

$$\text{Relative interface location} = (i - 1)/(L - 1)$$

where $i$ marks the residue at which half of the cumulative buried surface area of the interface is passed (i.e. interface midpoint), and $L$ is the sequence length.

The normalised distance between translational start points of two interfaces was calculated as:

$$\text{Relative translational distance} = \left( f_{\text{last}} - f_{\text{first}} \right) /L$$

where $f$ marks the first residue of the given interface and $L$ is the sequence length.

Area differences between the first and the last interfaces were normalised according to the equation:

$$\text{Relative size difference} = (BSA_{\text{first}} - BSA_{\text{last}})/(BSA_{\text{first}} + BSA_{\text{last}})$$

where BSA is the buried surface area of the corresponding interface.

## Calculation of interfacial contact-related properties

To make sure all software runs without errors, we converted the human structure files of homo- and heterodimers obtained by our pipeline from the Protein Data Bank to standard Brookhaven PDB format by taking the first atom locations in case of multiple AltLoc entries, converting non-canonical amino acids into equivalent standard names, renaming chain pairs with identical chain identifiers, and stripping files to only contain ATOM, TER, and END lines, thus excluding heteroatoms. Interfacial residue contacts were determined with the software PRODIGY (*Vangone and Bonvin, 2015*; *Xue et al., 2016*), using the default settings. The software RING 3.0 (*Clementel et al., 2022*) was used to compute the different types of residue interactions between the subunits. The network policy was set to 'closest' and all interactions were returned for a contact using the flag `--all_edges`.

## Determining protein isoelectric point from structure

Predicted structures of the human monomeric proteome were acquired from the AlphaFold database (*Tunyasuvunakool et al., 2021*) in PDB format, which were converted into PQR files with PDB2PQR (*Dolinsky et al., 2004*) using the PARSE force field. The PQR files were piped into the software BLUUES (*Walsh et al., 2012*), and the.ddg output was kept, containing the pH ~ charge titration data. Then, each protein's isoelectric point (pI) was calculated by interpolating from the pH ~ charge curve for charge = 0, using the approx() function in R. For proteins that are longer than 2700 amino acids and are contained in fragments in the AlphaFold database, we took the mean pI across the fragments.

## Protein localisation

We obtained annotations for plasma membrane, cytoplasmic, and nuclear localisations directly from the UniProt FTP site (*Bateman, 2021*). Canonical UniProt entries with the gene ontology terms plasma membrane (GO:0005886), cytoplasm (GO:0005737), and nucleus (GO:0005634) were considered.

## HEK293 active ribosome count

Normalised ribosome protected fragments of actively translating ribosomes specific to Human Embryonal Kidney 293 lineage were determined by *Clamer et al., 2018*, the data are available at the NCBI Gene Expression Omnibus (*Edgar et al., 2002*) under the accession GSE112353. We used averages from two biological replicates.

## Protein age

The ages of proteins were obtained from the work of *Liebeskind et al., 2016*, at the link https://github.com/marcottelab/Gene-Ages/tree/master/Main. The main_HUMAN and the main_YEAST comma separated value files were parsed and the modeage column was used in our analyses. We combined the mode age into two categories, where 'ancient' proteins are those whose genes are common to all cellular life ('Cellular_organisms'), whose genes were transferred horizontally from bacteria after eukaryotes diverged from archaea ('Euk +Bac'), and whose genes emerged in the clades of eukaryotes and archaea ('Eukaryota' and 'Euk_Archaea'). The other four age groups, ranging from genes emerged in the classes Opisthokonta to Mammalia in the human proteome, and from Ascomycota to Saccharomyceta in the yeast proteome, were classified as 'more recent'.

## hu.MAP2.0 heteromers

*Drew et al., 2021* integrated large-scale affinity purification mass spectrometry data sets, large-scale biochemical fractionation data, proximity labelling, and RNA hairpin pulldown data to generate a complex map with >7000 complexes, which is freely available at http://humap2.proteincomplexes.org/. We used hu.MAP2.0 to supplement genes of heteromers for the analysis in *Figure 4B*.

## Mapping simultaneously forming interfaces in multi-interface heteromeric subunits

Cotranslational assembly-onset positions were acquired from the supplemental material of *Bertolini et al., 2021*. From the onset positions, 30 residues were subtracted to account for the length of the ribosome tunnel. Our method maps the assembly-onset position to the closest interface midpoint in the linear sequence. Cases where the assembly-onset mapped to a homomeric interface were discarded under the assumption that the homomeric interface is hierarchically higher and undergoes *cis*-assembly. In subsequent analyses presented in *Figure 3B, C*, we only included comparisons between heteromeric interfaces.

## Mapping bacterial subunits to operons

Operon annotations were downloaded from OperomeDB (*Chetal and Janga, 2015*). Genes were mapped to UniProt identifiers using *E. coli* proteome-specific mapping from the UniProt FTP site (*Bateman, 2021*).

## Molecular graphics

Visualisation of structures was performed with UCSF ChimeraX version 1.1 (*Pettersen et al., 2021*).

## Statistical analysis

Data exploration and statistical analyses were carried out in RStudio (*Rstudio, 2022*) version 2022.02.0+443 'Prairie Trillium' Release, using R version 4.2.1 (*R Development Core Team, 2021*). The R packages used for analyses are tidyverse, tidytable, rsample, rstatix, scales, and ggbeeswarm. The Wilcoxon rank-sum or signed-rank tests were used for A/B testing of interface size distributions, because although they appear log-normal, they are also left bounded because of the minimum interface size cutoff, thus a non-parametric test was required. Wilcoxon signed-rank tests were one tailed, and their main assumption that the data are symmetric around the median was supported by boxplot

distributions. In Dunn's test of multiple comparisons the Holm–Bonferroni method (*Holm, 1979*) was used to correct for family wise error rate. The effect sizes were defined as the *Z*-score computed from the p value over the square root of sample size (*Tomczak and Tomczak, 2014*). In the bootstrap analyses, data were stratified for protein age or operonal localisation in $10^4$ resamples. The p value was calculated by determining the fraction of point estimates (difference in area) greater than 0, with correction for finite sampling (*Buckland et al., 1998*). In the regression analysis, conditions for statistical inference, including linearity of the relationship between variables, the independence and normality of the residuals, and homoscedasticity were met; validations can be found in the analysis scriptata and code availabilit.

## Acknowledgements

MB is supported by the Biotechnology and Biological Sciences Research Council EASTBIO DTP (BB/M010996/1), and used resources provided by the Edinburgh Compute and Data Facility. JAM was supported by a Medical Research Council Career Development Award (MR/M02122X/1) and is a Lister Institute Research Prize Fellow.

## Additional information

### Funding

| Funder | Grant reference number | Author |
| --- | --- | --- |
| Biotechnology and Biological Sciences Research Council | BB/M010996/1 | Mihaly Badonyi |
| Medical Research Council | MR/M02122X/1 | Joseph A Marsh |
| Lister Institute of Preventive Medicine | | Joseph A Marsh |

The funders had no role in study design, data collection, and interpretation, or the decision to submit the work for publication.

### Author contributions

Mihaly Badonyi, Conceptualization, Formal analysis, Writing - original draft; Joseph A Marsh, Conceptualization, Data curation, Supervision, Writing - review and editing

### Author ORCIDs

Mihaly Badonyi http://orcid.org/0000-0002-8305-5618
Joseph A Marsh http://orcid.org/0000-0003-4132-0628

### Decision letter and Author response

Decision letter https://doi.org/10.7554/eLife.79602.sa1
Author response https://doi.org/10.7554/eLife.79602.sa2

## Additional files

### Supplementary files
• MDAR checklist

### Data availability
Data and code to reproduce the results have been deposited on the OSF at https://osf.io/x5b2n/.

The following dataset was generated:

| Author(s) | Year | Dataset title | Dataset URL | Database and Identifier |
|---|---|---|---|---|
| Badonyi M | 2022 | Large protein complex interfaces have evolved to promote cotranslational assembly | https://osf.io/x5b2n/ | Open Science Framework, x5b2n |

The following previously published datasets were used:

| Author(s) | Year | Dataset title | Dataset URL | Database and Identifier |
|---|---|---|---|---|
| Humphreys IR | 2021 | Computed structures of core eukaryotic protein complexes | https://doi.org/10.5452/ma-bak-cepc | ModelArchive, 10.5452/ma-bak-cepc |
| Jumper JE | 2021 | *Homo sapiens* AlphaFold structures | https://alphafold.ebi.ac.uk/download | AlphaFold Database, Homo sapiens |
| Drew K, Wallingford JB, Marcotte EM | 2021 | hu.MAP 2.0: integration of over 15,000 proteomic experiments builds a global compendium of human multiprotein assemblies' | http://humap2.proteincomplexes.org/ | hu.MAP, 2.0 |
| Clamer M, Bernabò P, Lauria F, Tebaldi T, Viero G | 2018 | Active ribosome profiling with RiboLace and standard ribosome profiling in HEK-293 cells | https://www.ncbi.nlm.nih.gov/geo/query/acc.cgi?acc=GSE112353 | NCBI Gene Expression Omnibus, GSE112353 |
| Clamer M | 2017 | *Homo sapiens* (Human) | https://swissmodel.expasy.org/repository/species/9606 | SWISS-MODEL Repository, 9606 |
| Liebeskind BJ, McWhite CD, Marcotte EM | 2016 | Towards consensus gene ages | https://github.com/marcottelab/Gene-Ages | GitHub, marcottelab/Gene-Ages |

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
