## [Editor Report]

The authors use a combination of proteome-specific protein complex structures and publicly available ribosome profiling data to show that cotranslational assembly is favored by large N-terminal intermolecular interfaces. The manuscript represents an important contribution to the field of protein biosynthesis pathways by suggesting an intuitive evolutionary mechanism that can promote co-translational assembly pathways in mammalians, yeast, and bacteria.

---

## [Decision Letter]

**Decision letter after peer review:**

Thank you for submitting your article "Large protein complex interfaces have evolved to promote cotranslational assembly" for consideration by *eLife*. Your article has been reviewed by 4 peer reviewers, one of whom is a member of our Board of Reviewing Editors, and the evaluation has been overseen by Volker Dötsch as the Senior Editor. The reviewers have opted to remain anonymous.

Essential revisions:

1. The authors' main claim is that N-terminal interfaces are significantly larger than other interfaces and facilitate cotranslational assembly interactions. The authors show interface size differences of high confidence cotranslational assembly complexes compared to low confidence cotranslational assembly complexes compared to all others in Figure S1C, divided into symmetry groups. However, no statistical significance is provided. This is essential, as this provides the basis for the manuscript's main claim.

2. The manuscript states that the authors compared the areas of first and last translated interfaces in bacteria and yeast. However, no experimental nascent-proteome data is presented for bacteria and yeast. There have been several recent publications exploring co-translational complex assembly in both yeast and bacteria. The authors claim that large protein complex interfaces have evolved to promote cotranslational assembly could be greatly supported by utilizing some of these examples for verification.

3. Apart from the size of the interface and its location (N-term vs C-term), an important point to consider is protein-protein interaction potential (including hydrogen bonds, electrostatic and hydrophobic interactions, number of contacts, etc). Would these parameters also differ for cotranslationally assembling complexes vs posttranslationally assembling complexes and to what extent?

4. The manuscript states "However, since larger interfaces tend to cotranslationally assemble more frequently, we filtered the subunits to include cases where the size of at least one of the two interfaces is larger than 1000 or 2000 Å2". This is further demonstrated in Figure 3.

The reasoning for selecting these criteria is not clear. The authors should show the distribution of each group (all, 1000, 2000 Å2) of coco vs. non-coco, so one can see how likely is this assumption.

5. The authors show two examples of simultaneous cotranslational assembly between subunit pairs in heteromeric complexes: the subunits E and B1 of the V-type ATPase (pdb: 6wm2), and importin-9 with histone H2A (pdb: 6n1z) (Figure 2D). Structural data indicate that the V-type ATPase is highly dynamic with several conformations to this complex (Vasanthakumar, et al. (2022) Nat Struct Mol Biol), suggesting that the complex undergoes significant post-translational conformational rotations (see PDB: 7TMQ vs. 7TMM for example). The new rotational variants seem to challenge the prediction and calculation presented here. The authors should address these findings.

6. The effect size for how common it is for the N-terminus interface to be larger than the C-terminal interface should be computed in the following manner: Calculate the probability that for proteins that have two interfaces, the first interface will be larger than the second (relevant to Figure 3 and S2). This number should be reported in the abstract (as you highlight this find in the second to last sentence of the abstract), as well as the results and discussion. The reason for this is to help the community avoid over-generalizing the findings.

*Reviewer #1 (Recommendations for the authors):*

1. Statistical significance should be indicated in Figure S1C, as this is an essential comparison to the experimental data.

2. The approach presented in Figure 2A for mapping the interactions is not well described and Figure legend is missing details as to (i) why the assembly onset is shown at the time indicated (and the description of what the red element is and where this is taken from) and (ii) what are the blue, gray, and yellow areas. It would be also important to validate this method on a well-defined dataset.

*Reviewer #2 (Recommendations for the authors):*

The manuscript provides a clear analysis of current structural and experimental data, characterizing novel interface features facilitating cotranslational assembly as well as demonstrating their conservation.

The manuscript is an important contribution to the field of protein biogenesis, however, a few key points remain unanswered.

– The authors' main claim is that N` terminal interfaces are significantly larger than other interfaces and facilitate cotranslational assembly interactions. The authors show interface size differences of high confidence cotranslational assembly complexes compared to low confidence cotranslational assembly complexes compared to all others in Figure S1C, divided into symmetry groups.

However, no statistical significance is provided for any. This is required, as this provides the basis for the manuscript's main claim.

– In the manuscript it is suggested that "According to this theory, the largest interfaces in a complex correspond to the earliest forming subcomplexes within the assembly pathway, irrespective of the binding mode." This is shown for two examples, in Figure 2D: simultaneous cotranslational assembly between subunit pairs in heteromeric complexes: the subunits E and B1 of the V-type ATPase (pdb: 6wm2), and importin-9 with histone H2A (pdb: 6n1z).

However, for the V-type ATPase, there is new structural data from Vasanthakumar, et al. (2022) Nat Struct Mol Biol. Vasanthakumar and colleagues have shown the complex is highly dynamic, assigning several conformations to this complex, whether attached or not attached to the membrane as well as rotational variants. In one conformation the interface of subunits E is with A and in another with B. The results suggest the complex undergoes significant post-translational conformational rotations (see PDB: 7TMQ vs. 7TMM for example). The new rotational variants seem to challenge the prediction and calculation presented here. The authors should address these findings.

– The manuscript states that "We compared the areas of first and last translated interfaces in bacterial, yeast…". However, no experimental nascent-proteome data is presented for bacteria and yeast.

There have been several recent publications exploring co-translational complex assembly in both yeast and bacteria. For example, Kassem, et al., 2017 explored co-translational assembly of the SAGA complex in yeast; Lautier et al., Mol cell 2021 and Seidel et al., 2022 Nature Com; which combined structural with ribosome profiling and RIP-seq data to explore the co-translational assembly of the nuclear pore complex in yeast. The authors themselves cite several studies exploring co-translational assembly in yeast, including Duncan and Mata 2011; Shiber et al., 2018; Panasenko et al. 2019.

In bacteria, for example, Fujiwara et al., 2020, Cell reports, performed a nascent-proteome-wide study of co-translational interactions in *Bacillus subtilis*.

The authors' main claim is that large protein complex interfaces have evolved to promote cotranslational assembly, can be greatly supported by utilizing some of these examples for verification.

– "The manuscript states "However, since larger interfaces tend to cotranslationally assemble more frequently, we filtered the subunits to include cases where the size of at least one of the two interfaces is larger than 1000 or 2000 Å2". This is further demonstrated in Figure 3.

The reasoning for selecting these criteria is not clear. The authors should show the distribution of each group (all, 1000, 2000 Å2) of coco vs. non-coco, so one can see how likely is this assumption.

*Reviewer #3 (Recommendations for the authors):*

1. The authors found that cotranslationally assembling complexes have (on average) larger interfaces than those assumed to assemble after their complete synthesis. The authors considered the effect size of cotranslationally assembling complexes relative to posttranslationally assembling complexes with different types of symmetry and the nature of assembling subunits (e.g. home- and heterodimers, etc). Figure 1C however also shows that the interface size is generally larger in homomeric symmetry groups in comparison with e.g. heterodimers irrespective of the assembly mechanism. Could the authors comment on these differences and discuss them too?

2. Apart from the size of the interface and its location (N-term vs C-term), an important point to consider is protein-protein interaction potential (including hydrogen bonds, electrostatic and hydrophobic interactions, number of contacts, etc). Would these parameters also differ for cotranslationally assembling complexes vs posttranslationally assembling complexes and to what extent?

3. Ribosome surface properties and charge were suggested to impose certain limits on the nature of the cotranslationally forming surfaces. Specifically, electrostatic effects of the ribosomal surface on nascent polypeptide dynamics have been considered of substantial significance. In this regard, it would be important to answer the question of whether there is any charge bias in the cotranslationally assembling surface complexes vs posttranslationally assembling surface complexes that might help cotranslationally assembling complexes avoid unproductive interactions with the ribosome.

*Reviewer #4 (Recommendations for the authors):*

1. The authors need to calculate the p-values for Figure S1C. If they are not significant, what are we to make of that? It would seem the data does not support the relevant conclusion.

2. The authors should show all their results are robust to using the 'high confidence' only data.

3. The effect size for how common it is for the N-terminus interface to be larger than the C-terminal interface should be computed in the following manner: Calculate the probability that for proteins that have two interfaces, the first interface will be larger than the second (relevant to Figure 3 and S2). This number should be reported in the abstract (as you highlight this find in the second to last sentence of the abstract), as well as the results and discussion. The reason this is to help the community avoid over-generalizing the findings.

[Editors' note: further revisions were suggested prior to acceptance, as described below.]

Thank you for resubmitting your work entitled "Large protein complex interfaces have evolved to promote cotranslational assembly" for further consideration by *eLife*. Your revised article has been evaluated by Volker Dötsch (Senior Editor) and a Reviewing Editor.

The manuscript has been improved but there are some remaining issues that need to be addressed, as outlined below:

Please provide a full answer to the remaining criticism raised by referee 2. This is an essential revision. Please also take care to clarify the concern of referee 4, as it is your chance to ensure the impact of the paper.

*Reviewer #2 (Recommendations for the authors):*

The manuscript "Large protein complex interfaces have evolved to promote cotranslational assembly" by Mihaly Badonyi and Joseph A Marsh combines analysis of experimental data from ribosome profiling experiments (mainly Bertolini et al., 2021) with structural data on protein complexes to test the impact of interface size on co-translational complex assembly pathways in mammalians. The authors find a strong correlation between interface size and co-translational subunit association. Expanding their structural analysis to yeast and *E. coli* complexes, the authors find evidence supporting the hypothesis that large interfaces have evolved to promote co-translational assembly. Thus, larger N` terminal interfaces can serve to facilitate successful co-translational assembly interactions, protecting the nascent proteome.

The revised manuscript has greatly improved and the authors have addressed most of the major concerns.

The authors now provide p-values in the dot plot-boxplot chart in Figure 1 and figure supplement 1D; showing the statistical significance of interface size differences in co-translationally assembling homomers in humans. As this is the main claim of the manuscript, it was an essential revision.

The authors also provide a new aspect of the evolutionary complex's age vs co-translational propensity, as well as explore the other interfacial contact-based properties of the co-translational assembly.

However, one major concern has not been addressed by the authors in a clear manner. In regard to figure 3, heteromers analysis of the correlation between interface size and assembly order, relying on the assumption of stable interface formation. The authors present the example of the interface between the subunits E and B1 of the V-type ATPase (pdb: 6wm2); figure 3d. Vasanthakumar, et al. (2022) cryo-EM study, has shown, that the subunits E and B1 can dissociate post-translationally, following 120º rotation, where a novel interface between the same E subunit forms with a different B subunit (B2 and then B3, corresponding to rotational states 2, 3). See figure 1c of Vasanthakumar, et al., titled "Coordinated conformational changes in the V1 complex during V-ATPase reversible dissociation". In the PDBs mentioned in the authors' reply the same chain corresponding to the B1 subunit (chain B) forms interfaces with chain K (pdb:7tmq), then forms a new, very similar interface with chain I (pdb:7tmp), and in the other rotational state with chain G (pdb:7tmo). Thus, the example of a dynamic interface here does not support the authors' claim of stable interface formation.

*Reviewer #4 (Recommendations for the authors):*

The authors have addressed my concern regarding figure S1c. I am slightly concerned about their response to point 1, in which they basically say we can't restrict our analysis to high-confidence co-translational dimers because otherwise there is not enough data.

However, as long as other reviewers do not have serious concerns about this latter point, then I am fine with the manuscript as it is.

---

## [Author Response]

Essential revisions:1. The authors' main claim is that N-terminal interfaces are significantly larger than other interfaces and facilitate cotranslational assembly interactions. The authors show interface size differences of high confidence cotranslational assembly complexes compared to low confidence cotranslational assembly complexes compared to all others in Figure S1C, divided into symmetry groups. However, no statistical significance is provided. This is essential, as this provides the basis for the manuscript's main claim.

We agree that the confidence-based classification is an important part of the work reported by Bertolini et al., 2021, in which high-confidence candidates meet two conditions: their disome enrichment profile is impacted by puromycin and proteinase K, and the protein is either cytosolic or nuclear. The first condition tests how genuine the disomes are, because puromycin causes premature chain termination, and proteinase K cleaves interconnected nascent chains C-terminal to aliphatic and aromatic amino acids. Both of these treatments should dissolve true disomes. The second condition of the sole inclusion of cytosolic and nuclear proteins is due to an experimental limitation, namely, that for proteins localised to other compartments the team could not formally exclude the possibility that the ribosomes interact with membrane components of the translocation machinery and that is why they are found in the disome fraction.

Limiting our study to high confidence candidates is prohibitive to the structural analyses, in part because high confidence proteins only comprise 20% of all cotranslationally assembling candidates, and partly because the sample size further drops after mapping to the available structures of complexes. In an attempt to address this, we extended the human homomeric structure data with homology models from the SWISS-MODEL repository, which contains structures of homomeric complexes on condition that the interface is conserved (Bienert *et al.*, 2017; Waterhouse *et al.*, 2018), and was also used during the annotation of homomers in the study by Bertolini et al., 2021. We have updated the dotplot-boxplot chart in Figure 1 —figure supplement 1D to also show the mean interface size, and we calculated *p*-values comparing high confidence *versus* all other proteins (excluding low confidence). The differences between high confidence and all other subunits were found significant at symmetry level for all homomers, but not for heterodimers. There are no significant differences observed between high and low confidence proteins for any of the groups. Although we might expect that high confidence proteins should, on average, have larger interfaces than low confidence proteins (assuming that a greater fraction of them represent true cases of cotranslational assembly), we note that the size of the high confidence set is relatively small, especially when split by symmetry group, and that the average interface size of the high confidence proteins is higher for all homomers, except in the very small dihedral group. Overall, we think that the small differences between high and low confidence sets, as well as the small size of the high confidence set, justify our use of the combined sets throughout this study.

We have now added a new paragraph in the manuscript discussing these findings.

2. The manuscript states that the authors compared the areas of first and last translated interfaces in bacteria and yeast. However, no experimental nascent-proteome data is presented for bacteria and yeast. There have been several recent publications exploring co-translational complex assembly in both yeast and bacteria. The authors claim that large protein complex interfaces have evolved to promote cotranslational assembly could be greatly supported by utilizing some of these examples for verification.

We thank the reviewer for their comment as this was an excellent suggestion. We were familiar with the recommended literature, but we did not consider using their data because of their generally small scale. However, in the updated manuscript we collated instances of sequential assembly events of nascent chains and mature subunits in budding yeast (Table 1), which turned out to favour our hypothesis about the first *versus* last interface size difference being important for sequential cotranslational assembly. In this table, we could not consider Duncan and Mata, 2011, because it was performed in fission yeast, which has diverged from budding yeast more than 500 million years ago, lacks sufficient structural data in the PDB, and the mRNA co-immunoprecipitation method is not specific enough to be considered in our analysis. Also, we could not include Kassem, Villanyi and Collart, 2017, because the Spt20:Ada2 interaction has not been experimentally determined yet, nor has it been computed by Humphreys *et al.*, 2021.

In terms of bacteria, we are not aware of a large-scale data similar to that of Bertolini *et al.*, 2021 on cotranslational assembly. Although, Fujiwara *et al.*, 2020 present compelling results, their method represents a general survey of cotranslational interactions, including protein-nucleic acid binding, and is not specific enough to be considered in our analyses. The authors themselves only present one case study of the complex of SpoIIAB homodimer with the sporulation σ factor F, which does not contain a multi-interface heteromeric subunit.

The works of Lautier *et al.*, 2021 (yeast) and Seidel *et al.*, 2022 (human) are very exciting, but we could not identify structures of multi-interface nuclear pore complex subunits, because of a limited number of subcomplexes in the PDB. A full complex of the cytoplasmic ring of the *Xenopus laevis* nuclear pore has only just been created in its full detail (Fontana *et al.*, 2022), and the composite structure is not yet available (PDB: 7tdz, hold for release).

3. Apart from the size of the interface and its location (N-term vs C-term), an important point to consider is protein-protein interaction potential (including hydrogen bonds, electrostatic and hydrophobic interactions, number of contacts, etc). Would these parameters also differ for cotranslationally assembling complexes vs posttranslationally assembling complexes and to what extent?

We are grateful to the reviewer for this comment, because at the initial stages of the project, we put a lot of work into exploring other properties that might correlate with cotranslational assembly. Initially, we omitted these results, because we did not want to distract the reader from the main point, which is interface size. However, we strongly agree that it is important to consider protein interaction potential and other interfacial properties. Therefore, in the updated manuscript we include a new section titled Interface area is more important than other interfacial contact-based properties for explaining cotranslational assembly, presenting analyses of the interface properties of *C_2_* symmetric homodimers and heterodimers in cotranslational assembly. The reason for choosing these two groups is because they are well represented in the structural data set, and because they only have one interface, which makes it straightforward to draw conclusions from the results. We hope that the reviewers will find these results interesting, and that the community will appreciate their inclusion.

4. The manuscript states "However, since larger interfaces tend to cotranslationally assemble more frequently, we filtered the subunits to include cases where the size of at least one of the two interfaces is larger than 1000 or 2000 Å2". This is further demonstrated in Figure 3.The reasoning for selecting these criteria is not clear. The authors should show the distribution of each group (all, 1000, 2000 Å2) of coco vs. non-coco, so one can see how likely is this assumption.

Given our observation earlier in the manuscript that interfaces involved in cotranslational assembly tend to be large, then we expect that subunits that form only small interfaces will not undergo cotranslational assembly. Inclusion of these subunits in our analysis therefore reduces the observed first vs last interface size trend. Although the interface cutoffs may seem arbitrary, the actual values may not matter, because the importance is in the fact that larger interfaces have a higher propensity to cotranslationally assemble. When only subunits with interfaces >1000 and >2000 Å2 are considered, effectively excluding subunits with little chance of undergoing cotranslational assembly, we observe the N-terminal interface size bias is much stronger. We have attempted to make this point more clearly in manuscript text.

In Figure 1 —figure supplement 1D, we show the distribution of interface sizes for coco (simultaneous cotranslational assembly) vs non-coco (all other) proteins, for homomers and heterodimers, but we note that this is not directly comparable to the sizes of our cutoffs in our N *vs* C interface comparison, where we are comparing the sizes of two different interfaces from the same subunit. Nevertheless, they suggest that cutoffs on the order of 1000 or 2000 Å2, as we have chosen, should serve to enrich the fraction of cotranslationally assembly subunits in our analysis. We also show the full distribution of interface sizes for first and last interfaces in Figure 5 —figure supplement 1A, with the two thresholds highlighted.

To further support the first *versus* last interface size trend without using arbitrarily chosen interface size cutoffs, in the updated manuscript we have also now provided analyses relating to the evolutionary age of the subunits (Liebeskind, McWhite and Marcotte, 2016). We first highlight the correlation between subunit age and interface size (Figure 4A) and subunit age and cotranslational assembly as detected by Bertolini *et al.,* 2021 (Figure 4B), in a short new section called Evolutionarily more ancient subunits of complexes are more likely to undergo cotranslational assembly. We then use this data to classify multi-interface heteromeric subunits into “ancient” and “more recent” age groups to investigate if older proteins are more likely to have a larger first translated interface, which would provide a more direct evidence to our hypothesis. Indeed, we have found that the difference in size between the first and last interface strongly favours the first interface in both yeast and human to a significant extent (Figure 5D). Altogether, we believe these results well represent the main conclusion of our study, which is that large interfaces have evolved to promote cotranslational assembly, and make a prediction that given large-scale data on the sequential cotranslational assembly mode on heteromers, we should expect their first translated interfaces to be larger.

5. The authors show two examples of simultaneous cotranslational assembly between subunit pairs in heteromeric complexes: the subunits E and B1 of the V-type ATPase (pdb: 6wm2), and importin-9 with histone H2A (pdb: 6n1z) (Figure 2D). Structural data indicate that the V-type ATPase is highly dynamic with several conformations to this complex (Vasanthakumar, et al. (2022) Nat Struct Mol Biol), suggesting that the complex undergoes significant post-translational conformational rotations (see PDB: 7TMQ vs. 7TMM for example). The new rotational variants seem to challenge the prediction and calculation presented here. The authors should address these findings.

Both the E1 and B subunits of the V-type ATPase have been identified to cotranslationally assemble by Bertolini *et al.*, 2021, and the E1 subunit is a high confidence protein with a coiled-coil domain, a motif found enriched in cotranslational assembly. We looked at the structures of the V_1_ complex (7tmm, 7tmo, 7tmp, 7tmq) and those of the three states of the full V-type ATPase (7tmr, 7tms, 7tmt), which were deposited by Vasanthakumar *et al.*, 2022. Subunits E and B remain in contact in all conformations and states. We calculated the E:B interface midpoints in the linear sequence of subunit E in all seven structures, which resulted in residues 200 (7tmr), 206 (7tmo), and 209 (7tmm, 7tmp, 7tmq, 7tms, 7tmt). The yeast subunit E is 236 amino acids long, thus the 9-residue variation in interface midpoints represents only 3.8% of the total translated polypeptide length. These results suggest that despite the dynamic nature of the complex, using the interface midpoint is a robust and reliable way of mapping cotranslationally forming interfaces.

6. The effect size for how common it is for the N-terminus interface to be larger than the C-terminal interface should be computed in the following manner: Calculate the probability that for proteins that have two interfaces, the first interface will be larger than the second (relevant to Figure 3 and S2). This number should be reported in the abstract (as you highlight this find in the second to last sentence of the abstract), as well as the results and discussion. The reason for this is to help the community avoid over-generalizing the findings.

Although we think that simplifying this result to a single number could potentially obscure the interesting differences we observe between species and different types of complexes, we have now done this and added the following statement to the abstract: “When considering all together, we observe the N-terminal interface to be larger than the C-terminal interface 54% of the time, increasing to 64% when we exclude subunits with only small interfaces, which are unlikely to cotranslationally assemble.” We have also further expanded upon our discussion of these results with respect to different species in the text.

Reviewer #1 (Recommendations for the authors):1. Statistical significance should be indicated in Figure S1C, as this is an essential comparison to the experimental data.

Please see Essential revisions (1).

2. The approach presented in Figure 2A for mapping the interactions is not well described and Figure legend is missing details as to (i) why the assembly onset is shown at the time indicated (and the description of what the red element is and where this is taken from) and (ii) what are the blue, gray, and yellow areas. It would be also important to validate this method on a well-defined dataset.

We specified the source data in Methods, which is the work by Bertolini *et al.*, 2021, who developed a model based on a sigmoid fit to the monosome-to-disome enrichment profile with the aim of deriving the assembly-onsets for all candidates. We did include a paragraph in the previous Results section of the manuscript explaining how the assembly-onset is defined:

“To perform an analysis at the multi-interface level, we made use of the assembly-onset positions determined for every protein in the cotranslational assembly data set (Bertolini et al., 2021). An assembly-onset is a single residue in the protein sequence, whose codon is being decoded by the ribosome at the time of cotranslational assembly. In order to identify which interface the assembly-onsets belongs to, we mapped them to the closest interface midpoint in the linear protein sequence, as illustrated in Figure 2A. This is to avoid biases from large interfaces, which have many more interface residues and therefore a higher probability that an assembly-onset would map to them if the interface was not compressed into a single residue, the midpoint.”

We agree that it would be helpful to validate the method, but we could not identify multi-interface heteromeric subunits among the simultaneously assembling proteins discovered to date (Liu *et al.*, 2016; Kamenova *et al.*, 2019; Panasenko *et al.*, 2019) whose interfaces we mapped based on the assembly-onset positions, due to limitations of the structural data. We hope that future case studies will provide more mechanistic insight into the assembly of these subunits and make the validation of our method possible.

We thank the reviewer for noticing that the figure description was lacking. We made changes to specify that individual interfaces in the area chart are represented by different colours and improved the description of cumulative interface area.

Reviewer #2 (Recommendations for the authors):The manuscript provides a clear analysis of current structural and experimental data, characterizing novel interface features facilitating cotranslational assembly as well as demonstrating their conservation.The manuscript is an important contribution to the field of protein biogenesis, however, a few key points remain unanswered.– The authors' main claim is that N` terminal interfaces are significantly larger than other interfaces and facilitate cotranslational assembly interactions. The authors show interface size differences of high confidence cotranslational assembly complexes compared to low confidence cotranslational assembly complexes compared to all others in Figure S1C, divided into symmetry groups.However, no statistical significance is provided for any. This is required, as this provides the basis for the manuscript's main claim.

Please see Essential revisions (1).

– In the manuscript it is suggested that "According to this theory, the largest interfaces in a complex correspond to the earliest forming subcomplexes within the assembly pathway, irrespective of the binding mode." This is shown for two examples, in Figure 2D: simultaneous cotranslational assembly between subunit pairs in heteromeric complexes: the subunits E and B1 of the V-type ATPase (pdb: 6wm2), and importin-9 with histone H2A (pdb: 6n1z).However, for the V-type ATPase, there is new structural data from Vasanthakumar, et al. (2022) Nat Struct Mol Biol. Vasanthakumar and colleagues have shown the complex is highly dynamic, assigning several conformations to this complex, whether attached or not attached to the membrane as well as rotational variants. In one conformation the interface of subunits E is with A and in another with B. The results suggest the complex undergoes significant post-translational conformational rotations (see PDB: 7TMQ vs. 7TMM for example). The new rotational variants seem to challenge the prediction and calculation presented here. The authors should address these findings.

Please see Essential revisions (5).

– The manuscript states that "We compared the areas of first and last translated interfaces in bacterial, yeast…". However, no experimental nascent-proteome data is presented for bacteria and yeast.There have been several recent publications exploring co-translational complex assembly in both yeast and bacteria. For example, Kassem, et al., 2017 explored co-translational assembly of the SAGA complex in yeast; Lautier et al., Mol cell 2021 and Seidel et al., 2022 Nature Com; which combined structural with ribosome profiling and RIP-seq data to explore the co-translational assembly of the nuclear pore complex in yeast. The authors themselves cite several studies exploring co-translational assembly in yeast, including Duncan and Mata 2011; Shiber et al., 2018; Panasenko et al. 2019.In bacteria, for example, Fujiwara et al., 2020, Cell reports, performed a nascent-proteome-wide study of co-translational interactions in *Bacillus subtilis*.The authors' main claim is that large protein complex interfaces have evolved to promote cotranslational assembly, can be greatly supported by utilizing some of these examples for verification.

We thank the reviewer for these suggestions as it has helped us provide more supporting evidence for the first *versus* last interface size trend. Please see Essential Revisions (2).

– "The manuscript states "However, since larger interfaces tend to cotranslationally assemble more frequently, we filtered the subunits to include cases where the size of at least one of the two interfaces is larger than 1000 or 2000 Å2". This is further demonstrated in Figure 3.The reasoning for selecting these criteria is not clear. The authors should show the distribution of each group (all, 1000, 2000 Å2) of coco vs. non-coco, so one can see how likely is this assumption.

Please see Essential revisions (4).

Reviewer #3 (Recommendations for the authors):1. The authors found that cotranslationally assembling complexes have (on average) larger interfaces than those assumed to assemble after their complete synthesis. The authors considered the effect size of cotranslationally assembling complexes relative to posttranslationally assembling complexes with different types of symmetry and the nature of assembling subunits (e.g. home- and heterodimers, etc). Figure 1C however also shows that the interface size is generally larger in homomeric symmetry groups in comparison with e.g. heterodimers irrespective of the assembly mechanism. Could the authors comment on these differences and discuss them too?

While we agree that these differences are interesting on their own, they also represent fundamental properties of homomeric symmetry groups and heterodimers, which were reported before (see Table 2 in Jones and Thornton, 1996; and Figure S5 in Marsh and Teichmann, 2014). The main reason members of the cyclic and dihedral symmetry have larger subunit interfaces is because they have more than one interface. The difference in interface size between homo- and heterodimers has been speculated to be because isologous interfaces are generally more ancient (Kim *et al.*, 2006; Dayhoff *et al.*, 2010), and ancient proteins tend to have larger interfaces, as we show in Figure 4A in the updated manuscript. We have made changes to the second paragraph of the Results section to include these ideas for the reader.

2. Apart from the size of the interface and its location (N-term vs C-term), an important point to consider is protein-protein interaction potential (including hydrogen bonds, electrostatic and hydrophobic interactions, number of contacts, etc). Would these parameters also differ for cotranslationally assembling complexes vs posttranslationally assembling complexes and to what extent?

We thank the reviewer for their helpful suggestion, as it adds great value to the manuscript, and led to interesting observations. Please see Essential Revisions (3).

3. Ribosome surface properties and charge were suggested to impose certain limits on the nature of the cotranslationally forming surfaces. Specifically, electrostatic effects of the ribosomal surface on nascent polypeptide dynamics have been considered of substantial significance. In this regard, it would be important to answer the question of whether there is any charge bias in the cotranslationally assembling surface complexes vs posttranslationally assembling surface complexes that might help cotranslationally assembling complexes avoid unproductive interactions with the ribosome.

The reviewer raises an interesting point. It is without doubt that charge plays complex roles in protein biosynthesis, including the regulation of translation speed (Charneski and Hurst, 2013), and the recruitment of cotranslational chaperones (Döring *et al.*, 2017). There are also a number of theories concerned with effects attributed to ribosome surface properties (Knight *et al.*, 2013; Schavemaker, Śmigiel and Poolman, 2017; Cassaignau *et al.*, 2021; Deckert *et al.*, 2021). In the updated manuscript, we present new analyses (Figure 2) which will be of interest to the reviewer. We observe that cotranslationally assembling subunits of both *C_2_* symmetric homodimers and heterodimers have a larger fraction of charged residues on their surfaces, which tend to be more negative than positive. Our understanding is that it is positive residues in emerging nascent chains that interact with ribosome surfaces (Cassaignau *et al.*, 2021; Deckert *et al.*, 2021), which would suggest cotranslationally assembling subunits are not more likely to interact with the ribosome surface than other subunits. This is consistent with the finding that the role of these interactions is to halt cotranslational folding (Cassaignau *et al.*, 2021), but cotranslationally assembling subunits should fold cotranslationally to some extent to be able to present at least partially complete interfaces to their partners. We also include in the Results section four mutually non-exclusive hypotheses that we think might explain our observations with surface charge, one of which was kindly suggested by the reviewer.

Reviewer #4 (Recommendations for the authors):1. The authors need to calculate the p-values for Figure S1C. If they are not significant, what are we to make of that? It would seem the data does not support the relevant conclusion.

Please see Essential revisions (1).

2. The authors should show all their results are robust to using the 'high confidence' only data.

Please see Essential revisions (1).

3. The effect size for how common it is for the N-terminus interface to be larger than the C-terminal interface should be computed in the following manner: Calculate the probability that for proteins that have two interfaces, the first interface will be larger than the second (relevant to Figure 3 and S2). This number should be reported in the abstract (as you highlight this find in the second to last sentence of the abstract), as well as the results and discussion. The reason this is to help the community avoid over-generalizing the findings.

Please see Essential revisions (6).

References

Bergendahl, L. T. and Marsh, J. A. (2017) ‘Functional determinants of protein assembly into homomeric complexes’, *Scientific Reports*. Nature Publishing Group, 7(1). doi: 10.1038/s41598-017-05084-8.

Bertolini, M. *et al.* (2021) ‘Interactions between nascent proteins translated by adjacent ribosomes drive homomer assembly’, *Science*. American Association for the Advancement of Science, 371(6524), pp. 57–64. doi: 10.1126/science.abc7151.

Bienert, S. *et al.* (2017) ‘The SWISS-MODEL Repository-new features and functionality’, *Nucleic Acids Research*. Nucleic Acids Res, 45(D1), pp. D313–D319. doi: 10.1093/nar/gkw1132.

Cassaignau, A. M. E. *et al.* (2021) ‘Interactions between nascent proteins and the ribosome surface inhibit co-translational folding’, *Nature Chemistry*. Nature Publishing Group, 13(12), pp. 1214–1220. doi: 10.1038/s41557-021-00796-x.

Charneski, C. A. and Hurst, L. D. (2013) ‘Positively Charged Residues Are the Major Determinants of Ribosomal Velocity’, *PLoS Biology*. Public Library of Science, 11(3), p. e1001508. doi: 10.1371/journal.pbio.1001508.

Chen, X. and Mayr, C. (2022) ‘A working model for condensate RNA-binding proteins as matchmakers for protein complex assembly’, *RNA*. Cold Spring Harbor Laboratory Press, 28(1), pp. 76–87. doi: 10.1261/RNA.078995.121.

Dayhoff, J. E. *et al.* (2010) ‘Evolution of Protein Binding Modes in Homooligomers’, *Journal of Molecular Biology*. Academic Press, 395(4), pp. 860–870. doi: 10.1016/j.jmb.2009.10.052.

Deckert, A. *et al.* (2021) ‘Common sequence motifs of nascent chains engage the ribosome surface and trigger factor’, Proceedings of the National Academy of Sciences of the United States of America. National Academy of Sciences, 118(52). doi: 10.1073/pnas.2103015118.

Döring, K. *et al.* (2017) ‘Profiling Ssb-Nascent Chain Interactions Reveals Principles of Hsp70-Assisted Folding’, *Cell*, 170(2), pp. 298-311.e20. doi: 10.1016/j.cell.2017.06.038.

Duncan, C. D. S. and Mata, J. (2011) ‘Widespread cotranslational formation of protein complexes’, *PLoS Genetics*. Edited by D. A. Wolf, 7(12), p. e1002398. doi: 10.1371/journal.pgen.1002398.

Fontana, P. *et al.* (2022) ‘Structure of cytoplasmic ring of nuclear pore complex by integrative cryo-EM and AlphaFold.’, *Science (New York, N.Y.)*. American Association for the Advancement of Science, 376(6598), p. eabm9326. doi: 10.1126/science.abm9326.

Forrest, L. R. (2015) ‘Structural Symmetry in Membrane Proteins∗’, *Annual Review of Biophysics*. Annual Reviews Inc, 44(1), pp. 311–337. doi: 10.1146/annurev-biophys-051013-023008.

Fujiwara, K. *et al.* (2020) ‘Proteome-wide Capture of Co-translational Protein Dynamics in *Bacillus subtilis* Using TnDR, a Transposable Protein-Dynamics Reporter’, *Cell Reports*, 33(2). doi: 10.1016/j.celrep.2020.108250.

Goodsell, D. S. and Olson, A. J. (2000) ‘Structural symmetry and protein function’, Annual Review of Biophysics and Biomolecular Structure. Annual Reviews 4139 El Camino Way, P.O. Box 10139, Palo Alto, CA 94303-0139, USA, pp. 105–153. doi: 10.1146/annurev.biophys.29.1.105.

Halbach, A. *et al.* (2009) ‘Cotranslational assembly of the yeast SET1C histone methyltransferase complex’, *EMBO Journal*. John Wiley and Sons, Ltd, 28(19), pp. 2959–2970. doi: 10.1038/emboj.2009.240.

Humphreys, I. R. *et al.* (2021) ‘Computed structures of core eukaryotic protein complexes’, *Science*. American Association for the Advancement of Science (AAAS), 374(6573). doi: 10.1126/science.abm4805.

Jones, S. and Thornton, J. M. (1996) ‘Principles of protein-protein interactions’, Proceedings of the National Academy of Sciences of the United States of America. National Academy of Sciences, pp. 13–20. doi: 10.1073/pnas.93.1.13.

Kamenova, I. *et al.* (2019) ‘Co-translational assembly of mammalian nuclear multisubunit complexes’, *Nature Communications*. Nature Publishing Group, 10(1). doi: 10.1038/s41467-019-09749-y.

Kassem, S., Villanyi, Z. and Collart, M. A. (2017) ‘Not5-dependent co-translational assembly of Ada2 and Spt20 is essential for functional integrity of SAGA’, *Nucleic Acids Research*. Oxford Academic, 45(3), pp. 1186–1199. doi: 10.1093/NAR/GKW1059.

Keene, J. D. (2007) ‘RNA regulons: Coordination of post-transcriptional events’, *Nature Reviews Genetics*. Nature Publishing Group, pp. 533–543. doi: 10.1038/nrg2111.

Kim, W. K. *et al.* (2006) ‘The many faces of protein-protein interactions: A compendium of interface geometry’, *PLoS Computational Biology*. Public Library of Science, 2(9), pp. 1151–1164. doi: 10.1371/journal.pcbi.0020124.

Knight, A. M. *et al.* (2013) ‘Electrostatic effect of the ribosomal surface on nascent polypeptide dynamics’, *ACS Chemical Biology*, 8(6), pp. 1195–1204. doi: 10.1021/cb400030n.

Lautier, O. *et al.* (2021) ‘Co-translational assembly and localized translation of nucleoporins in nuclear pore complex biogenesis’, *Molecular cell*. Mol Cell, 81(11), pp. 2417-2427.e5. doi: 10.1016/J.MOLCEL.2021.03.030.

Levy, E. D. *et al.* (2008) ‘Assembly reflects evolution of protein complexes’, *Nature*. Nature Publishing Group, 453(7199), pp. 1262–1265. doi: 10.1038/nature06942.

Liebeskind, B. J., McWhite, C. D. and Marcotte, E. M. (2016) ‘Towards consensus gene ages’, *Genome Biology and Evolution*. Oxford Academic, 8(6), pp. 1812–1823. doi: 10.1093/gbe/evw113.

Liu, F. *et al.* (2016) ‘Cotranslational association of mRNA encoding subunits of heteromeric ion channels’, Proceedings of the National Academy of Sciences of the United States of America. National Academy of Sciences, 113(17), pp. 4859–4864. doi: 10.1073/pnas.1521577113.

Marsh, J. A. *et al.* (2013) ‘Protein complexes are under evolutionary selection to assemble via ordered pathways’, *Cell*, 153(2), pp. 461–470. doi: 10.1016/j.cell.2013.02.044.

Marsh, J. A. and Teichmann, S. A. (2014) ‘Protein Flexibility Facilitates Quaternary Structure Assembly and Evolution’, *PLoS Biology*. Edited by G. A. Petsko. Public Library of Science, 12(5), p. e1001870. doi: 10.1371/journal.pbio.1001870.

Panasenko, O. O. *et al.* (2019) ‘Co-translational assembly of proteasome subunits in NOT1-containing assemblysomes’, *Nature Structural and Molecular Biology*. Nature Publishing Group, 26(2), pp. 110–120. doi: 10.1038/s41594-018-0179-5.

Ponstingl, H. *et al.* (2005) ‘Morphological aspects of oligomeric protein structures’, *Progress in Biophysics and Molecular Biology*. Pergamon, pp. 9–35. doi: 10.1016/j.pbiomolbio.2004.07.010.

Schavemaker, P. E., Śmigiel, W. M. and Poolman, B. (2017) ‘Ribosome surface properties may impose limits on the nature of the cytoplasmic proteome’, *eLife*. *eLife* Sciences Publications Ltd, 6. doi: 10.7554/*ELIFE*.30084.

Seidel, M. *et al.* (2022) ‘Co-translational assembly orchestrates competing biogenesis pathways’, *Nature Communications*. Nature Publishing Group, 13(1), pp. 1–15. doi: 10.1038/s41467-022-28878-5.

Shiber, A. *et al.* (2018) ‘Cotranslational assembly of protein complexes in eukaryotes revealed by ribosome profiling’, *Nature*. Nature Publishing Group, 561(7722), pp. 268–272. doi: 10.1038/s41586-018-0462-y.

Vasanthakumar, T. *et al.* (2022) ‘Coordinated conformational changes in the V1 complex during V-ATPase reversible dissociation’, *Nature Structural and Molecular Biology 2022 29:5*. Nature Publishing Group, 29(5), pp. 430–439. doi: 10.1038/s41594-022-00757-z.

Waterhouse, A. *et al.* (2018) ‘SWISS-MODEL: Homology modelling of protein structures and complexes’, *Nucleic Acids Research*. Oxford Academic, 46(W1), pp. W296–W303. doi: 10.1093/nar/gky427.

Wells, J. N., Bergendahl, L. T. and Marsh, J. A. (2015) ‘Co-translational assembly of protein complexes’, *Biochemical Society Transactions*, 43, pp. 1221–1226. doi: 10.1042/BST20150159.

Wells, J. N., Bergendahl, L. T. and Marsh, J. A. (2016) ‘Operon Gene Order Is Optimized for Ordered Protein Complex Assembly’, *Cell Reports*, 14(4), pp. 679–685. doi: 10.1016/j.celrep.2015.12.085.

[Editors' note: further revisions were suggested prior to acceptance, as described below.]

The manuscript has been improved but there are some remaining issues that need to be addressed, as outlined below:Please provide a full answer to the remaining criticism raised by referee 2. This is an essential revision. Please also take care to clarify the concern of referee 4, as it is your chance to ensure the impact of the paper.Reviewer #2 (Recommendations for the authors):The manuscript "Large protein complex interfaces have evolved to promote cotranslational assembly" by Mihaly Badonyi and Joseph A Marsh combines analysis of experimental data from ribosome profiling experiments (mainly Bertolini et al., 2021) with structural data on protein complexes to test the impact of interface size on co-translational complex assembly pathways in mammalians. The authors find a strong correlation between interface size and co-translational subunit association. Expanding their structural analysis to yeast and *E. coli* complexes, the authors find evidence supporting the hypothesis that large interfaces have evolved to promote co-translational assembly. Thus, larger N` terminal interfaces can serve to facilitate successful co-translational assembly interactions, protecting the nascent proteome.The revised manuscript has greatly improved and the authors have addressed most of the major concerns.The authors now provide p-values in the dot plot-boxplot chart in Figure 1 and figure supplement 1D; showing the statistical significance of interface size differences in co-translationally assembling homomers in humans. As this is the main claim of the manuscript, it was an essential revision.The authors also provide a new aspect of the evolutionary complex's age vs co-translational propensity, as well as explore the other interfacial contact-based properties of the co-translational assembly.However, one major concern has not been addressed by the authors in a clear manner. In regard to figure 3, heteromers analysis of the correlation between interface size and assembly order, relying on the assumption of stable interface formation. The authors present the example of the interface between the subunits E and B1 of the V-type ATPase (pdb: 6wm2); figure 3d. Vasanthakumar, et al. (2022) cryo-EM study, has shown, that the subunits E and B1 can dissociate post-translationally, following 120º rotation, where a novel interface between the same E subunit forms with a different B subunit (B2 and then B3, corresponding to rotational states 2, 3). See figure 1c of Vasanthakumar, et al., titled "Coordinated conformational changes in the V1 complex during V-ATPase reversible dissociation". In the PDBs mentioned in the authors' reply the same chain corresponding to the B1 subunit (chain B) forms interfaces with chain K (pdb:7tmq), then forms a new, very similar interface with chain I (pdb:7tmp), and in the other rotational state with chain G (pdb:7tmo). Thus, the example of a dynamic interface here does not support the authors' claim of stable interface formation.

While there is evidence that subunit exchange may be widespread in proteomes (Tusk, Delalez and Berry, 2018), it was suggested by the authors of the study that cotranslationally assembled components are the least likely to exchange. Vasanthakumar *et al.*, 2022, demonstrate large conformational changes throughout the V_1_ subcomplex on separation from the V_O_ subcomplex in the yeast V-ATPase. Their work represents an important contribution to understanding how the regulator of the ATPase of vacuoles and endosomes (RAVE) complex might reassemble V_1_ and V_O_ by recruiting subunit C, as well as revealing the inherent dynamism of the complex in mechanistic detail. However, subunits B and E do not change relative positions or undergo subunit exchange during the conformational rearrangements described by the authors. The apparent distinct interfaces formed by the E subunit with different B subunits in the different states are due to differences in author chain name assignment between the different PDB files. The reviewer correctly notes that chain B directly interacts with chains G, I and K in the three different structures, 7tmo, 7tmp and 7tmq, respectively. However, chains G, I and K in these different structures are in fact referring to the same subunit, *i.e.* there is not really a ~120° rotation between the E subunits in the different states 1, 2 and 3, the authors have simply named the chains differently in the different files. This can be clearly observed in the video created by the authors to illustrate the conformational changes that occur during the cycle of dissociation and reassembly of the V_1_ and V_O_ subcomplexes (Video 4, https://www.nature.com/articles/s41594-022-00757-z#Sec22). The video shows how the two subcomplexes dissociate during glucose starvation, whereby the a and d subunits (green and cyan, respectively) first initiate the mechanical inhibition of proton conductance through V_O_ (Kishikawa *et al.*, 2020), which is followed by the V_1_ subcomplex adopting an inhibited conformation as well (state 2). The latter requires the ~150° rotation of the C-terminal region of subunit H (H_CT_, orange) towards the central rotor and the subsequent dissociation of subunit C (dark orange). Following these steps, the V_1_ subcomplex, now devoid of subunit C, is allowed to transition between states 1, 2, and 3, despite lacking magnesium ATPase activity. However, the interconversion between states is not associated with exchange of the B or E subunits (red and purple, respectively), and each pair of interacting B and E subunits remains in direct contact across all states. The mechanistic events reported by Vasanthakumar *et al.* are therefore consistent with cotranslational assembly of subunits B and E.

We would also like to note that conformational rearrangements post-assembly are not incompatible with the cotranslational assembly of subunits, since proteins with large interfaces are inherently more flexible (Marsh and Teichmann, 2014) and thus more amenable to conformational changes, functional or otherwise. An example of this is subunits of the proteasome, which have been observed to cotranslationally assemble by multiple independent studies (Duncan and Mata, 2011; Shiber *et al.*, 2018; Panasenko *et al.*, 2019). Strikingly, 19 out of 32 subunits in the human 26S proteasome (pdb: 6msd) were found to cotranslationally assemble by Bertolini *et al.*, with 8 out of 19 being high confidence. The work by Dambacher *et al.,* 2016, illustrates well the dynamic rearrangements during lid incorporation into the 26S proteasome (Video 1), whereas the work by Dong *et al.*, 2018, shows the complete dynamic process of the substrate engagement (Video 1). Thus, while random dissociation of cotranslationally assembled components remains unlikely, dynamic conformational changes thereof are not disfavoured.

We have now updated the text to note that while major conformational rearrangements can occur, the same B and E subunits remain in contact across all states, and mention the potential for large posttranslational rearrangements in structures being common in cotranslationally assembling subunits.

Reviewer #4 (Recommendations for the authors):The authors have addressed my concern regarding figure S1c. I am slightly concerned about their response to point 1, in which they basically say we can't restrict our analysis to high-confidence co-translational dimers because otherwise there is not enough data.However, as long as other reviewers do not have serious concerns about this latter point, then I am fine with the manuscript as it is.

We acknowledge the concern of the reviewer, and that our analysis is inevitably limited by the available complexes in the PDB. We hope that the growing intensity of experimental structure determination and improving computational approaches will resolve the remaining complexes in the very near future, making possible further advances upon structural studies such as ours. We would like to emphasise that the analysis presented in Figure S1A strongly suggests that there is no artificial interface size bias in response to biochemical fractionation, which justifies the use of pooled high and low confidence dimers for further use. Additionally, the first *versus* last interface size analysis, essentially the main conclusion of the study, does not hinge upon the results obtained via analysis of the Bertolini data.

References

Bertolini, M. *et al.* (2021) ‘Interactions between nascent proteins translated by adjacent ribosomes drive homomer assembly’, *Science*. American Association for the Advancement of Science, 371(6524), pp. 57–64. doi: 10.1126/science.abc7151.

Dambacher, C. M. *et al.* (2016) ‘Atomic structure of the 26S proteasome lid reveals the mechanism of deubiquitinase inhibition’, *eLife*. *eLife* Sciences Publications Ltd, 5(JANUARY2016). doi: 10.7554/*eLife*.13027.

Dong, Y. *et al.* (2019) ‘Cryo-EM structures and dynamics of substrate-engaged human 26S proteasome’, *Nature*. Nature Publishing Group, 565(7737), pp. 49–55. doi: 10.1038/s41586-018-0736-4.

Duncan, C. D. S. and Mata, J. (2011) ‘Widespread cotranslational formation of protein complexes’, *PLoS Genetics*. Edited by D. A. Wolf, 7(12), p. e1002398. doi: 10.1371/journal.pgen.1002398.

Kishikawa, J. I. *et al.* (2020) ‘Mechanical inhibition of isolated Vo from V/A-Atpase for proton conductance’, *eLife*. *eLife* Sciences Publications Ltd, 9, pp. 1–20. doi: 10.7554/*eLife*.56862.

Marsh, J. A. and Teichmann, S. A. (2014) ‘Protein Flexibility Facilitates Quaternary Structure Assembly and Evolution’, *PLoS Biology*. Edited by G. A. Petsko. Public Library of Science, 12(5), p. e1001870. doi: 10.1371/journal.pbio.1001870.

Panasenko, O. O. *et al.* (2019) ‘Co-translational assembly of proteasome subunits in NOT1-containing assemblysomes’, *Nature Structural and Molecular Biology*. Nature Publishing Group, 26(2), pp. 110–120. doi: 10.1038/s41594-018-0179-5.

Shiber, A. *et al.* (2018) ‘Cotranslational assembly of protein complexes in eukaryotes revealed by ribosome profiling’, *Nature*. Nature Publishing Group, 561(7722), pp. 268–272. doi: 10.1038/s41586-018-0462-y.

Tusk, S. E., Delalez, N. J. and Berry, R. M. (2018) ‘Subunit Exchange in Protein Complexes’, *Journal of Molecular Biology*. Academic Press, pp. 4557–4579. doi: 10.1016/j.jmb.2018.06.039.

Vasanthakumar, T. *et al.* (2022) ‘Coordinated conformational changes in the V1 complex during V-ATPase reversible dissociation’, *Nature Structural and Molecular Biology 2022 29:5*. Nature Publishing Group, 29(5), pp. 430–439. doi: 10.1038/s41594-022-00757-z.